# INDETERMINATE PROBABILITY THEORY

## ABSTRACT

Currently, there is no mathematical analytical form for a general posterior. We have discovered a new theory to address this issue, which is called Indeterminate Probability Theory. This is a big discovery in the field of probability, and it is an extension of classical probability theory, and makes classical probability theory a special case to our theory. In this paper, we propose a new perspective for understanding probability theory by introducing Observers and treating the outcome of each random experiment as an indeterminate probability distribution, which leads to probability calculations being a combination of ground truth and observation errors. We then discover three conditional mutual independent assumptions as **Candidate Axioms** and divide the probability process into two phase: observation phase and inference phase. In the observation phase, a general equation for any complex posterior is derived. In the inference phase, the inference probability equation with the posterior is derived. Base on this theory, we propose a new general model called IPNN – Indeterminate Probability Neural Network to validate our theory. Furthermore, in one of our another papers, this new theory is successfully applied to the task of multivariate time series (MTS) forecasting without relying on any neural models, and it outperforms LSTM models as well as some transformer-based models. Anonymous (2024b) In addition, further applications of this new theory are also discussed in this paper. Validations of this theory are reflected in experimental results.

## 1 INTRODUCTION

Currently, for a general posterior $P\left(Y = y_l \mid A^1 = a_{i_1}^1, \ldots, A^N = a_{i_N}^N\right)$ or more compactly written as $P\left(y_l | a_{i_1}^1, a_{i_2}^2, \ldots, a_{i_N}^N\right)$[1], the status of analytical solutions are as below:

Table 1: Reading Symbols

| Symbol | Description |
|---|---|
| $X = x_k$ | $k^{th}$ random experiment, $k = 1, 2, \ldots, n$ |
| $Y = y_l$ | event of random variable $Y, l = 1, 2, \ldots, m$ |
| $A^j = a_{i_j}^j$ | event of random variable $A^j, i_j = 1, 2, \ldots, M_j, j = 1, 2, \ldots, N$ |
| $\mathbb{A} = (A^1, A^2, \ldots, A^N)$ | joint (multivariate) random variables |

**General Probability Form**

- **Equation**:

$$\frac{\text{number of event } (Y = y_l, A^1 = a_{i_1}^1, \ldots, A^N = a_{i_N}^N) \text{ occurs}}{\text{number of event } (A^1 = a_{i_1}^1, \ldots, A^N = a_{i_N}^N) \text{ occurs}} \tag{1}$$

- **Assumption**: No assumption.

---

[1] Most of probabilities are formulated compactly in this paper.

- **Limitations**:
  1. Not applicable if $A^j$ is continuous.
  2. Not applicable for indeterminate case.
  3. Joint sample space is exponentially large.

- **Space Size**: $m \cdot \prod_{j=1}^{N} M_j$

**Naïve Bayes Form**

- **Equation**:

$$\frac{P(Y = y_l) \cdot \prod_{j=1}^{N} P(A^j = a_{i_j}^j \mid Y = y_l)}{P(A^1 = a_{i_1}^1, \ldots, A^N = a_{i_N}^N)} \tag{2}$$

- **Assumption**: Given $Y$, $A^1, A^2, \ldots, A^N$ conditionally independent.
- **Limitations**:
  1. Assumption is strong.
  2. $P(A^j = a_{i_j}^j \mid Y = y_l)$ is not always solvable.

- **Space Size**: $m \cdot \sum_{j=1}^{N} M_j$

**Indeterminate Probability Form**

- **Equation**: Equation (10)
- **Assumption**: Given $X$, $A^1, A^2, \ldots, A^N$ and $Y$ conditionally independent. see Candidate Axiom 1 and Candidate Axiom 2.
- **Limitations**: No. (Joint sample space is exponentially large only when Monte Carlo method is not used.)
- **Space Size**: $m \cdot n \cdot N \cdot C$ (or $m \cdot \prod_{j=1}^{N} M_j$ without Monte Carlo method, see Section 3.4.)

Due to the limitations of general probability form and Naïve Bayes form, MCMC Robert & Casella (2004) and variational inference methods Jordan et al. (1999) as approximate solutions are well developed in the past.

In this paper, we propose a new probability theory as the analytical solution to address the aforementioned limitations.

## 2 BACKGROUND

Let's first introduce a small game – coin toss: a child and an adult are observing the outcomes of each coin toss and record the results independently (heads or tails), the child can't always record the results correctly and the adult can record it correctly, in addition, the records of the child are also observed by the adult. After several coin tosses, the question now is, suppose the adult is not allowed to watch the next coin toss, what is the probability of his inference outcome of next coin toss via the child's record?

As shown in Figure 1, random variables X is the random experiment itself, and $X = x_k$ represent the $k^{th}$ random experiment. Y and A are defined to represent the adult's record and the child's record, respectively. And $hd, tl$ is for heads and tails. For example, after 10 coin tosses, the records are shown in Table 2.

We formulate X compactly with the ground truth, as shown in Table 3.

Through the adult's record Y and the child's records A, we can calculate $P(Y|A)$, as shown in Table 4. We define this process as observation phase.

For next coin toss ($X = x_{11}$), the question of this game is formulated as calculation of the probability $P^A(Y|X)$, superscript A indicates that Y is inferred via record A, not directly observed by the adult. For example, given the next coin toss $X = hd = x_{11}$, the child's record has then two situations: $P(A = hd|X = hd = x_{11}) = 4/5$ and $P(A = tl|X = hd = x_{11}) = 1/5$. With the adult's

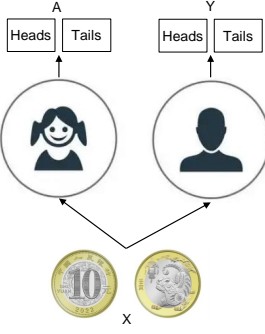

Figure 1: Example of coin toss game.

Table 2: Example of 10 times coin toss outcomes

| Experiment | Truth | A | Y |
|---|---|---|---|
| $X = x_1$ | $hd$ | $A = hd$ | $Y = hd$ |
| $X = x_2$ | $hd$ | $A = hd$ | $Y = hd$ |
| $X = x_3$ | $hd$ | $A = hd$ | $Y = hd$ |
| $X = x_4$ | $hd$ | $A = hd$ | $Y = hd$ |
| $X = x_5$ | $hd$ | $\boldsymbol{A = tl}$ | $Y = hd$ |
| $X = x_6$ | $tl$ | $A = tl$ | $Y = tl$ |
| $X = x_7$ | $tl$ | $A = tl$ | $Y = tl$ |
| $X = x_8$ | $tl$ | $A = tl$ | $Y = tl$ |
| $X = x_9$ | $tl$ | $A = tl$ | $Y = tl$ |
| $X = x_{10}$ | $tl$ | $A = tl$ | $Y = tl$ |
| $X = x_{11}$ | $hd$ | $A = ?$ | $Y = ?$ |

Table 3: The adult's and child's records: $P(Y|X)$ and $P(A|X)$

| $\frac{\#(Y,X)}{\#(X)}$ | $Y = hd$ | $Y = tl$ | $\frac{\#(A,X)}{\#(X)}$ | $A = hd$ | $A = tl$ |
|---|---|---|---|---|---|
| $X = hd$ | 5/5 | 0 | $X = hd$ | 4/5 | 1/5 |
| $X = tl$ | 0 | 5/5 | $X = tl$ | 0 | 5/5 |

observation of the child's records, we have $P(Y = hd|A = hd) = 4/4$ and $P(Y = hd|A = tl) = 1/6$. Therefore, given next coin toss $X = hd = x_{11}$, $P^A(Y = hd|X = hd = x_{11})$ is the summation of these two situations: $\frac{4}{5} \cdot \frac{4}{4} + \frac{1}{5} \cdot \frac{1}{6}$. Table 4 answers the above mentioned question.

Table 4: Results of observation and inference phase: $P(Y|A)$ and $P^A(Y|X)$

| $\frac{\#(Y,A)}{\#(A)}$ | $Y = hd$ | $Y = tl$ | $\sum_A \left( \frac{\#(A,X)}{\#X} \cdot \frac{\#(Y,A)}{\#A} \right)$ | $Y = hd$ | $Y = tl$ |
|---|---|---|---|---|---|
| $A = hd$ | 4/4 | 0 | $X = hd = x_{11}$ | $\frac{4}{5} \cdot \frac{4}{4} + \frac{1}{5} \cdot \frac{1}{6}$ | $\frac{4}{5} \cdot 0 + \frac{1}{5} \cdot \frac{5}{6}$ |
| $A = tl$ | 1/6 | 5/6 | $X = tl = x_{11}$ | $0 \cdot \frac{4}{4} + \frac{5}{5} \cdot \frac{1}{6}$ | $0 \cdot 0 + \frac{5}{5} \cdot \frac{5}{6}$ |

Let's go one step further, we can find that even the child's record is written in unknown language (e.g. $A \in \{ZHENG, FAN\}$), Table 4 can still be calculated by the man. The same is true if the child's record is written from the perspective of attributes, such as color, shape, etc.

The most important difference between classical probability theory and indeterminate probability theory is that indeterminate probability theory introduces an Observer to observe the outcome of random experiments. For this example, the child and adult act as two different observers, which leads to different records for the same random experiments. This is the core idea of Indeterminate Probability Theory.

## 3 INDETERMINATE PROBABILITY THEORY

### 3.1 DEFINITION OF INDETERMINATE PROBABILITY

Define a special random variable $X$ for the random experiments, and $X = x_k$ is for $k^{th}$ experiment. $A^1, A^2, ..., A^N$ and $Y$ are different random variables, details see Table 1. The following equation is always true:

$$P(x_k) \equiv \frac{1}{n}, k = 1, 2, \ldots, n. \tag{3}$$

Actually, we must inevitably use an observer (e.g. a machine, a model, a human, etc.) to observe the outcome of random experiments, and we may get a probability distribution estimation for $k^{th}$ experiment. (This part is not focused in the past.)

Therefore, indeterminate probability is for indicating the observed outcome of $k^{th}$ experiment as random variables $A^j$ (or $Y$), which can be mathematically represented as

$$\text{Indeterminate Probability} := P\left(a_{i_j}^j \mid x_k\right) \in [0, 1] \tag{4}$$

In classical probability theory, the event state $A^j = a_{i_j}^j$ for $k^{th}$ experiment (or the outcome of $k^{th}$ experiment for random variable $A^j$) has only two situations: happened or not happened, which leads to the indeterminate probability $P\left(A^j = a_{i_j}^j \mid X = x_k\right) \in \{0, 1\}$. For example, in Section 2, the third coin toss outcome is head, this is represented as $P\left(Y = hd \mid X = x_3\right) = 1$ and $P\left(Y = tl \mid X = x_3\right) = 0$.

This difference makes the general probability Equation (1) not applicable anymore.

Besides, the observers also bring observation errors to the probability calculation results, more details to understand this point, see another coin toss example in Appendix B.

For multivariate variables $\mathbb{A} = \left(A^1, A^2, \ldots, A^N\right)$, the observations from different observers are independent [2] , we find

**Candidate Axiom 1.** *Given $X$, $A^1, A^2, \ldots, A^N$ are conditionally mutually independent.*

And according to Candidate Axiom 1, the joint indeterminate probability is

$$P\left(a_{i_1}^1, a_{i_2}^2, \ldots, a_{i_N}^N \mid x_k\right) = \prod_{j=1}^N P\left(a_{i_j}^j \mid x_k\right) \in [0, 1] \tag{5}$$

Where it can be easily proved,

$$\sum_{\mathbb{A}} \prod_{j=1}^N P\left(a_{i_j}^j \mid x_k\right) = 1, k = 1, 2, \ldots, n. \tag{6}$$

In classical probability, the joint indeterminate probability $\prod_{j=1}^N P\left(a_{i_j}^j \mid x_k\right) \in \{0, 1\}$.

### 3.2 OBSERVATION PHASE

In observation phase, the relationship between all random variables $A^1, A^2, \ldots, A^N$ and $Y$ is established after the whole observations, the posterior is formulated as:

$$P\left(y_l \mid a_{i_1}^1, a_{i_2}^2, \ldots, a_{i_N}^N\right) = \frac{P\left(y_l, a_{i_1}^1, a_{i_2}^2, \ldots, a_{i_N}^N\right)}{P\left(a_{i_1}^1, a_{i_2}^2, \ldots, a_{i_N}^N\right)} \tag{7}$$

Because the general probability Equation (1) is not applicable for indeterminate case, the joint probability is calculated according to total probability theorem over all samples $X = (x_1, x_2, \ldots, x_n)$, and with Equation (5) and Equation (3) we have:

$$
\begin{aligned}
P\left(a_{i_1}^1, a_{i_2}^2, \ldots, a_{i_N}^N\right) &= \sum_{k=1}^n \left(P\left(a_{i_1}^1, a_{i_2}^2, \ldots, a_{i_N}^N \mid x_k\right) \cdot P(x_k)\right) \\
&= \sum_{k=1}^n \left(\prod_{j=1}^N P\left(a_{i_j}^j \mid x_k\right) \cdot P(x_k)\right) = \frac{\sum_{k=1}^n \left(\prod_{j=1}^N P\left(a_{i_j}^j \mid x_k\right)\right)}{n}
\end{aligned}
\tag{8}
$$

---

[2]Empirically, we find that Candidate Axiom 1 and Candidate Axiom 2 are also true for the same observer. $Y$ and $A^1, A^2, ..., A^N$, are understood as the same observer with difference perspectives.

Because $Y = y_l$ and $A^j = a_{i_j}^j$ also comes from different observers (or same observer with different perspectives), we find

**Candidate Axiom 2.** *Given $X$, $A^j$ and $Y$ are conditionally mutually independent in the observation phase, $j = 1, 2, \ldots, N$.*

Therefore, according to total probability theorem, Equation (5), Equation (3) and Candidate Axiom 2, we derive

$$
\begin{aligned}
P\left(y_l, a_{i_1}^1, a_{i_2}^2, \ldots, a_{i_N}^N\right) &= \sum_{k=1}^{n}\left(P\left(y_l, a_{i_1}^1, a_{i_2}^2, \ldots, a_{i_N}^N \mid x_k\right) \cdot P(x_k)\right) \\
&= \sum_{k=1}^{n}\left(P\left(y_l \mid x_k\right) \cdot \prod_{j=1}^{N} P\left(a_{i_j}^j \mid x_k\right) \cdot P(x_k)\right) \\
&= \frac{\sum_{k=1}^{n}\left(P\left(y_l \mid x_k\right) \cdot \prod_{j=1}^{N} P\left(a_{i_j}^j \mid x_k\right)\right)}{n}
\end{aligned}
\tag{9}
$$

Substitute Equation (8) and Equation (9) into Equation (7), we have:

$$
P\left(y_l \mid a_{i_1}^1, a_{i_2}^2, \ldots, a_{i_N}^N\right) = \frac{\sum_{k=1}^{n}\left(P\left(y_l \mid x_k\right) \cdot \prod_{j=1}^{N} P\left(a_{i_j}^j \mid x_k\right)\right)}{\sum_{k=1}^{n}\left(\prod_{j=1}^{N} P\left(a_{i_j}^j \mid x_k\right)\right)}
\tag{10}
$$

Where it can be proved,

$$
\sum_{l=1}^{m} P\left(y_l \mid a_{i_1}^1, a_{i_2}^2, \ldots, a_{i_N}^N\right) = 1
\tag{11}
$$

Equation (10) is the analytical solution for any general posterior. If $P(a_{i_j}^j \mid x_k) \in \{0, 1\}$ and $P(y_l \mid x_k) \in \{0, 1\}$, this equation is mathematically equivalent to the general probability Equation (1).

### 3.3 INFERENCE PHASE

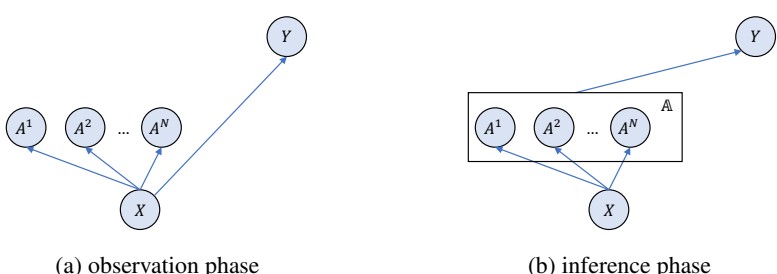

(a) observation phase          (b) inference phase

Figure 2: Independence illustration with Bayesian network.

Given $\mathbb{A}$, with Equation (10) (passed experience) $Y = y_l$ can be inferred, this inferred $y_l$ has no pointing to any specific sample $x_k$, incl. also new input sample $x_{n+1}$. We find

**Candidate Axiom 3.** *Given $\left(A^1, A^2, \ldots, A^N\right)$, $X$ and $Y$ are conditionally mutually independent in the inference phase.*

The difference between observation and inference phase is that we do not have the outcome of $Y = y_l$ for new experiment $X = x_{n+1}$ during inference phase, we can reasonably divide the probability process into these two phases. Otherwise, Candidate Axiom 2 and Candidate Axiom 3 cannot both be true.

Therefore, for next experiment $X = x_{n+1}$, according to total probability theorem over joint sample space $\left(a_{i_1}^1, a_{i_2}^2, \ldots, a_{i_N}^N\right) \in \mathbb{A}$, with Candidate Axiom 3, Equation (5) and Equation (10), we have the inference probability with the posterior as

$$
\begin{aligned}
P^{\mathbb{A}}\left(y_l \mid x_{n+1}\right) &= \sum_{\mathbb{A}}\left(P\left(y_l, a_{i_1}^1, a_{i_2}^2, \ldots, a_{i_N}^N \mid x_{n+1}\right)\right) \\
&= \sum_{\mathbb{A}}\left(P\left(y_l \mid a_{i_1}^1, a_{i_2}^2, \ldots, a_{i_N}^N\right) \cdot P\left(a_{i_1}^1, a_{i_2}^2, \ldots, a_{i_N}^N \mid x_{n+1}\right)\right) \\
&= \sum_{\mathbb{A}}\left(\frac{\sum_{k=1}^n\left(P\left(y_l \mid x_k\right) \cdot \prod_{j=1}^N P\left(a_{i_j}^j \mid x_k\right)\right)}{\sum_{k=1}^n\left(\prod_{j=1}^N P\left(a_{i_j}^j \mid x_k\right)\right)} \cdot \prod_{j=1}^N P\left(a_{i_j}^j \mid x_{n+1}\right)\right)
\end{aligned}
\tag{12}
$$

Where superscript $\mathbb{A}$ indicates that the inference is based on the latent variables $\mathbb{A}$. $P^{\mathbb{A}}\left(y_l \mid x_{n+1}\right)$ and $P\left(y_l \mid x_k\right)$ are mathematically the same thing. The first one represents the indeterminate probability for the inferred outcome, while the latter one represents the indeterminate probability for the observed outcome.

The estimated inference outcome for discrete decision making is

$$
\hat{y} := \underset{l \in \{1,2,\ldots,m\}}{\arg\max} \; P^{\mathbb{A}}\left(y_l \mid x_{n+1}\right)
\tag{13}
$$

### 3.4 COMPLEXITY REDUCTION

According to the idea proposed in Anonymous (2024a), due to the constraint in Equation (6), Equation (12) can be expressed in the form of an expectation. Consequently, we can utilize the Monte Carlo method to approximate it, thereby transforming the problem from exponential complexity to polynomial time and space complexity, that is from $m \cdot \prod_{j=1}^N M_j$ to $m \cdot n \cdot N \cdot C$.

$$
\begin{aligned}
P^{\mathbb{A}}\left(y_l \mid x_{n+1}\right) &= \mathbb{E}_{a_{i_j}^j \sim P\left(a_{i_j}^j \mid x_{n+1}\right)}\left[\frac{\sum_{k=1}^n\left(P\left(y_l \mid x_k\right) \cdot \prod_{j=1}^N P\left(a_{i_j}^j \mid x_k\right)\right)}{\sum_{k=1}^n\left(\prod_{j=1}^N P\left(a_{i_j}^j \mid x_k\right)\right)}\right] \\
&\approx \frac{1}{C}\sum_{c=1}^C\left(\frac{\sum_{k=1}^n\left(P\left(y_l \mid x_k\right) \cdot \prod_{j=1}^N P\left(a_{i_j,c}^j \mid x_k\right)\right)}{\sum_{k=1}^n\left(\prod_{j=1}^N P\left(a_{i_j,c}^j \mid x_k\right)\right)}\right),
\end{aligned}
\tag{14}
$$

where $a_{i_j,c}^j \sim P\left(a_{i_j}^j \mid x_{n+1}\right)$ and C is for Monte Carlo number.

Unlike MCMC Robert & Casella (2004), which requires a large number of samples from a complex and large space. In CIPNN, we only need sample two points ($C = 2$), even for a 1000-dimensional latent space. For more details, please refer to CIPNN Anonymous (2024a).

### 3.5 SUMMARY

Our most important contribution is that we propose a new general **analytical** and **tractable** probability Equation (12), rewritten as:

$$
\boldsymbol{P^{\mathbb{A}}\left(Y = y_l \mid X = x_{n+1}\right) =}
$$

$$
\sum_{\mathbb{A}}\left(\underbrace{\frac{\sum_{k=1}^n\left(\boldsymbol{P\left(Y = y_l \mid X = x_k\right)} \cdot \prod_{j=1}^N P\left(A^j = a_{i_j}^j \mid X = x_k\right)\right)}{\sum_{k=1}^n\left(\prod_{j=1}^N P\left(A^j = a_{i_j}^j \mid X = x_k\right)\right)}}_{\text{Observation phase}} \cdot \prod_{j=1}^N P\left(A^j = a_{i_j}^j \mid X = x_{n+1}\right)\right)
$$

$$
\underbrace{\phantom{\sum_{\mathbb{A}}}}_{\text{Inference phase}}
$$

$$
\tag{15}
$$

Where $X = x_k$ denote the $k^{th}$ random experiment, $Y$ and $A^{1:N}$ are different discrete or continuous Anonymous (2024a;b) random variables.

Our proposed theory is derived from three our proposed conditional mutual independent assumptions, see Candidate Axiom 1, Candidate Axiom 2 and Candidate Axiom 3. However, in our opinion, these axioms can neither be proved nor falsified, and we do not find any exceptions until now. Since this theory cannot be mathematically proved, we can only validate it through experiment.

The three Candidate Axioms are the basis of our theory, therefore, we suggest that readers try to find counterexamples to these axioms. Even if a toy dataset is found that contradicts these axioms, the validity of our proposed theory shall be significantly diminished.

Finally, our proposed indeterminate probability theory is an extension of classical probability theory, and classical probability theory is one special case to our theory. More details to understand our theory intuitively, see Appendix A.

## 4 APPLICATIONS

### 4.1 IPNN

For neural network tasks, $X = x_k$ is for the $k^{th}$ input sample, $P(y_l|x_k) = y_l(k) \in [0, 1]$ is for the soft/hard label of train sample $x_k$, $P^{\mathbb{A}}(y_l \mid x_t)$ is for the predicted label of test sample $x_t$.

Figure 3 shows IPNN model architecture, the output neurons of a general neural network (FFN, CNN, Resnet He et al. (2016), Transformer Vaswani et al. (2017), Pretrained-Models Devlin et al. (2019), etc.) is split into N unequal/equal parts, the split shape is marked as Equation (16), hence, the number of output neurons is the summation of the split shape $\sum_{j=1}^{N} M_j$. Next, each split part is passed to 'softmax', so the output neurons can be defined as discrete random variable $A^j \in \left\{ a_1^j, a_2^j, \ldots, a_{M_j}^j \right\}, j = 1, 2, \ldots, N$, and each neuron in $A^j$ is regarded as an event. After that, all the random variables together form the N-dimensional joint sample space, marked as $\mathbb{A} = (A^1, A^2, \ldots, A^N)$, and all the joint sample points are fully connected with all labels $Y \in \{y_1, y_2, \ldots, y_m\}$ via conditional probability $P\left(y_l|a_{i_1}^1, a_{i_2}^2, \ldots, a_{i_N}^N\right)$.

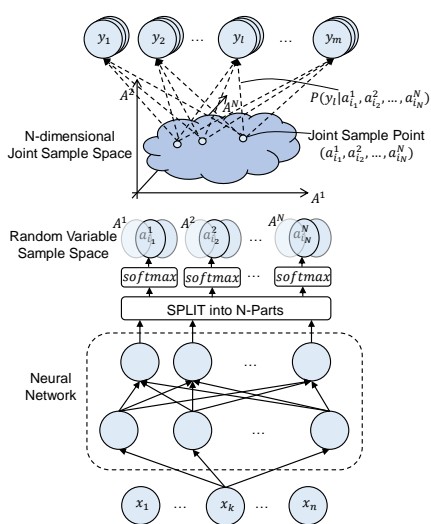

Figure 3: IPNN model architecture. $P\left(y_l|a_{i_1}^1, a_{i_2}^2, \ldots, a_{i_N}^N\right)$ is statistically calculated, not model weights.

$$\text{Split shape} := \{M_1, M_2, \ldots, M_N\} \quad (16)$$

Given an input sample $x_k$, let $\alpha_{i_j}^j(k)$ be the model outputted value after 'softmax'. With Assumption 1, the indeterminate probability (model output) is

$$P\left(a_{i_j}^j \mid x_k\right) := \alpha_{i_j}^j(k) \quad (17)$$

**Assumption 1.** *For neural networks, given an input sample $X = x_k$, **IF** $\sum_{i_j=1}^{M_j} \alpha_{i_j}^j(k) = 1$ and $\alpha_{i_j}^j(k) \in [0, 1], k = 1, 2, \ldots, n$. **THEN**, $\left\{a_1^j, a_2^j, \ldots, a_{M_j}^j\right\}$ can be regarded as collectively exhaustive and exclusive events set, they are partitions of the sample space of random variable $A^j, j = 1, 2, \ldots, N$.*

According to Equation (12), the prediction for test sample $x_t$ is

$$P^{\mathbb{A}}\left(y_l \mid x_t\right) = \sum_{\mathbb{A}} \left( \frac{\sum_{k=1}^{n} \left( y_l(k) \cdot \prod_{j=1}^{N} \alpha_{i_j}^{j}(k) \right)}{\sum_{k=1}^{n} \left( \prod_{j=1}^{N} \alpha_{i_j}^{j}(k) \right)} \cdot \prod_{j=1}^{N} \alpha_{i_j}^{j}(t) \right) \qquad (18)$$

We use cross entropy as loss function:

$$\mathcal{L} = -\sum_{l=1}^{m} \left( y_l(k) \cdot \log P^{\mathbb{A}}\left(y_l \mid x_t\right) \right) \qquad (19)$$

More details on IPNN, including the introduction, related work, training Strategy, limitations, etc., can be found in Appendix E.

### 4.2 CIPNN AND CIPAE

In Anonymous (2024a), we extended the indeterminate probability distribution to continuous random variable distribution. We propose a general classification model called CIPNN, which works even for a 1000-dimensional latent space.

Besides, we propose a general auto-encoder called CIPAE, which do not even have the decoder component. The framework between CIPAE and VAE Kingma & Welling (2014) is almost the same, but VAE must use a neural network as the decoder. This is a special ability of our analytical solution.

### 4.3 MTS FORECASTING

In Anonymous (2024b), it shows how to consider multivariate point value as indeterminate probability distribution. And the multivariate time series (MTS) forecasting problem is formulated as a complex posterior without relying on any neural models, and the method even does not need any training process. With our proposed theory, the complex posterior becomes analytical tractable, even in the presence of a thousand-dimensional latent space.

Although our proposed theory is motivated by design of new neural network architectures, it is not limited to neural networks. This is supported by our MTS forecasting method, which serves as strong evidence.

## 5 VALIDATIONS

The validations in this section are focusing on our proposed Candidate Axioms. More validations or usefulness of our theory, you can also find in Anonymous (2024a;b).

### 5.1 EVALUATION ON DATASETS

Results on MNIST Deng (2012), Fashion-MNIST Xiao et al. (2017), CIFAR10 Krizhevsky et al. (2009) and STL10 Coates et al. (2011) show that our proposed indeterminate probability theory is valid, the backbone between IPNN, CIPNN and 'Simple-Softmax' is the same, the last layer of the latter one is connected to softmax function. Although IPNN and CIPNN does not reach any SOTA, the results are very important evidences to our proposed mutual independence assumptions, see Candidate Axiom 1, Candidate Axiom 2 and Candidate Axiom 3.

Table 5: Test accuracy with 3-D latent space; backbone is FCN for MNIST and Fashion-MNIST, Resnet50 He et al. (2016) for CIFAR10 and STL10.

| Dataset | CIPNN | IPNN | Simple-Softmax |
|---|---|---|---|
| MNIST | $95.9 \pm 0.3$ | $95.8 \pm 0.5$ | $97.6 \pm 0.2$ |
| Fashion-MNIST | $85.4 \pm 0.3$ | $84.5 \pm 1.0$ | $87.8 \pm 0.2$ |
| CIFAR10 | $81.3 \pm 1.6$ | $83.6 \pm 0.5$ | $85.7 \pm 0.9$ |
| STL10 | $92.4 \pm 0.4$ | $91.6 \pm 4.0$ | $94.7 \pm 0.7$ |

## 5.2 EVALUATION ON LARGE LATENT SPACE

For IPNN, we cannot use Monte Carlo method to reduce the exponential complexity (Section 3.4), otherwise, IPNN will be not able to do back-propagation. Hence, we validate IPNN till to 20-D dimension.

Besides, for larger latent space, IPNN has also over-fitting problem, train accuracy is 98.7% for 10-D space and 99.5% for 20-D space, this is only the limitation of IPNN, not CIPNN.

Table 6: Average test accuracy of 10 times results on Large Latent Space on MNIST.

| Latent space | 5-D | 10-D | 20-D | 50-D | 100-D | 200-D | 500-D | 1000-D |
|---|---|---|---|---|---|---|---|---|
| IPNN | 94.8 | 88.6 | 80.6 | - | - | - | - | - |
| CIPNN | 95.6 | 94.7 | 94.7 | 94.9 | 94.9 | 94.9 | 94.7 | 93.4 (2 times) |

## 5.3 EVALUATION WITH DUPLICATED RANDOM VARIABLE INFERENCE

If the latent variables are the same, i.e., $A^1$ is identical to $A^2$, then this is the most critical case for Candidate Axiom 1.

In Appendix B.1, we use a coin toss example to show that Candidate Axiom 1 works for this simple example. Besides, in Anonymous (2024b), we have duplicated the MTS dataset for abuse test of our theory, and results show that it has no negative effect to the forecasting performance.

## 6 CONCLUSION

Since we now consider the state of event in an indeterminate way, we have opened the door to the applicability of indeterminate probability theory in various fields:

For instance, similar to MTS forecasting we can also interpret a point from data clusters as indeterminate probability, then we can do supervised classification task. We can interpret the outputs of multi-models as indeterminate probability, then we can do ensemble learning related task. These applications are also not neural models. Even in the field of physics, with our limited understanding of the 'Uncertainty Principle' Britannica (2023), we can interpret the position of particles as indeterminate probability distribution, similar to MTS forecasting Anonymous (2024b), and then do some inference or forecasting tasks.

Definitively, our proposed indeterminate theory is not limited to the applications discussed in this paper. More applications of this theory are worth to be researched in the future.

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

## A    An Intuitive Explanation

Since our proposed indeterminate probability theory is quite new, we will explain this idea by comparing it with classical probability theory, see below table:

Table 7: An intuitive comparison between classical probability theory and our proposed theory.

| | |
|---|---|
| Observation (Classical) | $P\left(Y = y_l \mid A^j = a^j_{i_j}\right) = \frac{\text{number of event } (Y=y_l, A^j=a^j_{i_j}) \text{ occurs}}{\text{number of event } (A^j=a^j_{i_j}) \text{ occurs}}$ |
| Inference (Classical) | $X = x_{n+1} \xrightarrow[\textbf{Determinate}]{P\left(A^j=a^j_{i_j}\mid X=x_{n+1}\right)=1} A^j = a^j_{i_j} \xrightarrow[\text{infer}]{P\left(Y=y_l\mid A^j=a^j_{i_j}\right)} Y = y_l$ |
| Observation (Ours) | $P\left(Y = y_l \mid A^j = a^j_{i_j}\right) = \frac{\text{sum of event } (Y=y_l, A^j=a^j_{i_j}) \text{ occurs, in decimal}}{\text{sum of event } (A^j=a^j_{i_j}) \text{ occurs, in decimal}}$ |
| Inference (Ours) | $X = x_{n+1} \begin{cases} \xrightarrow{P\left(A^j=a^j_1\mid X=x_{n+1}\right)\in[0,1]} & A^j = a^j_1 & \xrightarrow{P\left(Y=y_l\mid A^j=a^j_1\right)} \\ \xrightarrow{P\left(A^j=a^j_2\mid X=x_{n+1}\right)\in[0,1]} & A^j = a^j_2 & \xrightarrow{P\left(Y=y_l\mid A^j=a^j_2\right)} \\ \xrightarrow{\cdots} & A^j = \ldots & \xrightarrow{\cdots} \\ \xrightarrow[\textbf{Indeterminate}]{P\left(A^j=a^j_{M_j}\mid X=x_{n+1}\right)\in[0,1]} & A^j = a^j_{M_j} & \xrightarrow[\text{infer}]{P\left(Y=y_l\mid A^j=a^j_{M_j}\right)} \end{cases} Y = y_l$ |

Note: Replacing $A^j$ with joint random variable $(A^1, A^2, \ldots, A^N)$ is also valid for above explanation.

In other word, for classical probability theory, perform a random experiment $X = x_k$, the event state is Determinate (happened or not happened), the probability is calculated by counting the number of occurrences, we define this process here as observation phase. For inference, perform a new random experiment $X = x_{n+1}$, the state of $A^j = a^j_{i_j}$ is Determinate again, so condition on $X = x_{n+1}$ is equivalent to condition on $A^j = a^j_{i_j}$, that may be the reason why condition on $X = x_{n+1}$ is not discussed explicitly in the past.

However, for our proposed indeterminate probability theory, perform a random experiment $X = x_k$, the event state is Indeterminate (understood as partly occurs), the probability is calculated by summing the decimal value of occurrences in observation phase. For inference, perform a new random experiment $X = x_{n+1}$, the state of $A^j = a^j_{i_j}$ is Indeterminate again, each case contributes the inference of $Y = y_l$, so the inference shall be the summation of all cases. Therefore, condition on $X = x_{n+1}$ is now different with condition on $A^j = a^j_{i_j}$, we need to explicitly formulate it, see Equation (15).

Once again, our proposed indeterminate probability theory does not have any conflict with classical probability theory, the observation and inference phase of classical probability theory is one special case to our theory.

## B    Example of Continuous Indeterminate Probability

This section is a copy from CIPNN. Anonymous (2024a)

We will use a simple coin toss example to demonstrate how to use Equation (10) and Eq. Equation (12) for continuous random variables, see Table 8.

**Observer**$_1$    Let's say, Observer$_1$ is an adult and record the outcome of coin toss always correctly, so the probability of $Y$ can be easily calculated with the general probability form:

$$P(Y = hd) = \frac{\text{number of } (Y = hd) \text{ occurs}}{\text{number of random experiments}} = \frac{5}{10} \tag{20}$$

Table 8: Example of coin toss.

| Random Experiment ID $X$ | $x_1$ | $x_2$ | $x_3$ | $x_4$ | $x_5$ |
| | $x_6$ | $x_7$ | $x_8$ | $x_9$ | $x_{10}$ |
|---|---|---|---|---|---|
| Ground Truth | $hd$ | $hd$ | $hd$ | $hd$ | $hd$ |
| | $tl$ | $tl$ | $tl$ | $tl$ | $tl$ |
| Record of Observer$_1$ $Y$ | $hd$ | $hd$ | $hd$ | $hd$ | $hd$ |
| | $tl$ | $tl$ | $tl$ | $tl$ | $tl$ |
| Equivalent Record $Y$ | 1, 0 | 1, 0 | 1, 0 | 1, 0 | 1, 0 |
| | 0, 1 | 0, 1 | 0, 1 | 0, 1 | 0, 1 |
| Record of Observer$_2$ $A$ | 0.8, 0.2 | 0.7, 0.3 | 0.9, 0.1 | 0.6, 0.4 | 0.8, 0.2 |
| | 0.1, 0.9 | 0.2, 0.8 | 0.3, 0.7 | 0.1, 0.9 | 0.2, 0.8 |
| Record of Observer$_3$ $z$ | $\mathcal{N}(3,1)$ | $\mathcal{N}(3,1)$ | $\mathcal{N}(3,1)$ | $\mathcal{N}(3,1)$ | $\mathcal{N}(3,1)$ |
| | $\mathcal{N}(-3,1)$ | $\mathcal{N}(-3,1)$ | $\mathcal{N}(-3,1)$ | $\mathcal{N}(-3,1)$ | $\mathcal{N}(-3,1)$ |

Where $hd$ is for head, $tl$ is for tail. And condition on $x_k$ is the indeterminate probability, e.g. $P(Y = hd|X = x_3) = 1$, $P(A = tl|X = x_6) = 0.9$ and $P(z|X = x_8) = \mathcal{N}(z; -3, 1)$.

If we represent Observer$_1$'s record with equivalent form of $P(Y = hd|X = x_k)$, the probability is:

$$P(Y = hd) = \sum_{k=1}^{10} P(Y = hd|X = x_k) \cdot P(X = x_k) = \frac{5}{10} \tag{21}$$

**Observer$_2$**  Let's say, Observer$_2$ is a model, it takes the image of each coin toss outcome as inputs, and it's outputs are discrete probability distribution.

The Observer$_2$'s record probability is

$$P(A = hd) = \sum_{k=1}^{10} P(A = hd|X = x_k) \cdot P(X = x_k) = \frac{4.7}{10} \tag{22}$$

This calculation result is a combination of **ground truth** and **observation errors**.

**Observer$_3$**  Let's say, Observer$_3$ is a strange unknown observer, it always outputs a Gaussian distribution for each coin toss with a 'to-be-discovered' pattern. How can we find this pattern?

$$P(z) = \sum_{k=1}^{10} P(z|X = x_k) \cdot P(X = x_k) = \frac{5 \cdot \mathcal{N}(z; 3, 1) + 5 \cdot \mathcal{N}(z; -3, 1)}{10} \tag{23}$$

We get a complexer $P(z)$ distribution here, it's form is still analytical. And this distribution have two bumps, how can we know the representation of each bump mathematically? We need to use the Observer$_1$'s record $Y$. With Equation (10) we have

$$P(Y = hd|z) = \frac{\sum_{k=1}^{10} P(Y = hd|X = x_k) \cdot P(z|X = x_k)}{\sum_{k=1}^{10} P(z|X = x_k)} = \frac{\mathcal{N}(z; 3, 1)}{\mathcal{N}(z; 3, 1) + \mathcal{N}(z; -3, 1)} \tag{24}$$

For next coin toss, let $P(z|X = x_{11}) = \mathcal{N}(z; 3, 1)$, With Equation (12) and Monte Carlo method, we have

$$P^z(Y = hd|X = x_{11}) = \int_z \left( P(Y = hd|z) \cdot P(z|X = x_{11}) \right)$$

$$= \mathbb{E}_{z \sim P(z|X=x_{11})} \left[ P(Y = hd|z) \right] \approx \frac{1}{C} \sum_{c=1}^{C} P(Y = hd|z_c) \quad (25)$$

$$= \frac{1}{C} \sum_{c=1}^{C} \frac{\mathcal{N}(z_c; 3, 1)}{\mathcal{N}(z_c; 3, 1) + \mathcal{N}(z_c; -3, 1)} \approx 1, z_c \sim \mathcal{N}(z; 3, 1)$$

Where $C$ is for Monte Carlo number. In this way, we know that the bump with mean value 3 is for $Y = hd$. Note: this issue cannot be analytically solved with current other probability theories.

If we use a neural network to act as observer$_3$ to output multivariate Gaussian distributions, this is the core idea of our CIPNN and CIPAE model, and their forms are still analytical.

### B.1 EXAMPLE OF DUPLICATED RANDOM VARIABLES INFERENCE

This section is a most critical example to Candidate Axiom 1.

Let $\mathbf{z} = (z, z, ...)^N$, we use N same random variable z for the inference, with Equation (10) we have

$$P(Y = hd|z, z, ...) = \frac{\sum_{k=1}^{10} P(Y = hd|X = x_k) \cdot P(z|X = x_k)^N}{\sum_{k=1}^{10} P(z|X = x_k)^N}$$

$$= \frac{\mathcal{N}(z; 3, 1)^N}{\mathcal{N}(z; 3, 1)^N + \mathcal{N}(z; -3, 1)^N} \quad (26)$$

For next coin toss, let $P(z|X = x_{11}) = \mathcal{N}(z; 3, 1)$, with Equation (12), similar to Equation (25), we have

$$P^{\mathbf{z}}(Y = hd|X = x_{11}) = \frac{1}{C} \sum_{c=1}^{C} \frac{\mathcal{N}(z_c; 3, 1)^N}{\mathcal{N}(z_c; 3, 1)^N + \mathcal{N}(z_c; -3, 1)^N} \approx 1, z_c \sim \mathcal{N}(z; 3, 1) \quad (27)$$

We can see that even for duplicated random variables, our calculation results are also almost not effected.

## C PROPERTIES OF INDETERMINATE PROBABILITY THEORY

The indeterminate probability theory (see Equation (15)) may have the following properties, some have not been proved mathematically due to our limited knowledge.

**Proposition 1.** *IF given A, B and Y is independent, we have $P(Y \mid A, B) = P(Y \mid A)$, **THEN**:*

$$P^{(A,B)}(Y \mid X = x_{n+1}) = P^A(Y \mid X = x_{n+1}) \quad (28)$$

*This property is understood as: Suppose given A, B and Y is independent, so B does not contribute for the inference.*

*Proof.*

$$
\begin{aligned}
&P^{(A,B)}\left(Y \mid X = x_{n+1}\right) \\
&= \sum_{A,B} \left(P\left(Y \mid A, B\right) \cdot P\left(A, B \mid X = x_{n+1}\right)\right) \\
&= \sum_{A,B} \left(P\left(Y \mid A\right) \cdot P\left(A \mid X = x_{n+1}\right) \cdot P\left(B \mid X = x_{n+1}\right)\right) \\
&= \sum_{A} \left(P\left(Y \mid A\right) \cdot P\left(A \mid X = x_{n+1}\right)\right) \cdot \sum_{B} P\left(B \mid X = x_{n+1}\right) \\
&= \sum_{A} \left(P\left(Y \mid A\right) \cdot P\left(A \mid X = x_{n+1}\right)\right) \\
&= P^A\left(Y \mid X = x_{n+1}\right)
\end{aligned}
\tag{29}
$$

$\square$

**Hypothesis 1.** *Let $Y, V$ be any two different random variables, Similarly, according to Candidate Axiom 1, we have $P(Y, V \mid X = x_{n+1}) = P(Y \mid X = x_{n+1}) \cdot P(V \mid X = x_{n+1})$. Our hypothesis is:*

$$
P^A\left(Y, V \mid X = x_{n+1}\right) = P^A\left(Y \mid X = x_{n+1}\right) \cdot P^A\left(V \mid X = x_{n+1}\right)
\tag{30}
$$

*This property is understood as: Given $X$, $Y$ and $V$ is independent, so the inference outcome is also independent.*

**Hypothesis 2.** *Let $P\left(A \mid X = x_{n+1}\right) \in [0, 1)$ and*

$$
\begin{aligned}
P\left(Y^0 = y_l \mid X = x_{n+1}\right) &= P^A\left(Y = y_l \mid X = x_{n+1}\right) \\
P\left(Y^1 = y_l \mid X = x_{n+1}\right) &= P^{Y^0}\left(Y = y_l \mid X = x_{n+1}\right) \\
P\left(Y^2 = y_l \mid X = x_{n+1}\right) &= P^{Y^1}\left(Y = y_l \mid X = x_{n+1}\right) \\
&\cdots
\end{aligned}
\tag{31}
$$

*Our hypothesis is:*

$$
P^{Y^\infty}\left(Y = y_l \mid X = x_{n+1}\right) = \frac{1}{m}, l = 1, 2, \ldots, m.
\tag{32}
$$

*This property is understood as: The inference accuracy will become poor as the information is transmitted one after another (from $Y^{i-1}$ to $Y^i$).*

**Hypothesis 3.** *Let $P\left(Y = y_l \mid X = x_{n+1}\right) \in \{0, 1\}$ and $P\left(A \mid X = x_{n+1}\right) \in [0, 1)$. Our hypothesis is:*

$$
\max_{l=1,2,\ldots,m} P^{(A,A)}\left(Y = y_l \mid X = x_{n+1}\right) > \max_{l=1,2,\ldots,m} P^{(A)}\left(Y = y_l \mid X = x_{n+1}\right)
\tag{33}
$$

*This property is understood as: The inference tendency will get more stronger with more same information $(A, A)$.*

## D   WHY IS INDETERMINATE PROBABILITY THEORY IS GOOD?

Table 9: Comparison of independence assumptions

|  | Assumption | Validity | Assumption Range |
|---|---|---|---|
| Example | $A^1, \ldots, A^N$ independent | Strongest assumption | all samples |
| Naïve Bayes | Given $Y$, $A^1, \ldots, A^N$ independent | Strong assumption | few samples |
| Ours | See our Candidate Axioms. | No exception | one sample |

Let's think the independent assumption in another way. Sometimes, $A^1, A^2, \ldots, A^N$ independence assumption is strong. Nevertheless, in the case of Naïve Bayes, the whole samples are partitioned into

small groups due to condition on $Y = y_l$, the conditional independence maybe not strong anymore. This maybe the reason why Naïve Bayes is successful for many applications.

For our proposed Candidate Axioms, the whole samples are partitioned into a single sample due to $X = x_k$, our assumptions are the most weak one. For example, even if $A^1$ is identical to $A^2$, our independent assumptions still hold true. Furthermore, we have already conducted tests with thousand of latent variables in CIPNN, these assumptions have proven to remain valid. In IPNN, you can test with a few variables due to the exponentially large space size during the training phase, but not during the prediction phase (Monte Carlo).

# E  IPNN

## E.1  INTRODUCTION

Humans can distinguish at least 30,000 basic object categories Biederman (1987), classification of all these would have two challenges: It requires huge well-labeled images; Model with softmax for large scaled datasets is computationally expensive. Zero-Shot Learning – ZSL Lampert et al. (2009); Fu et al. (2018) method provides an idea for solving the first problem, which is an attribute-based classification method. ZSL performs object detection based on a human-specified high-level description of the target object instead of training images, like shape, color or even geographic information. But labelling of attributes still needs great efforts and expert experience. Hierarchical softmax can solve the computationally expensive problem, but the performance degrades as the number of classes increase Mohammed & Umaashankar (2018).

Probability theory has not only achieved great successes in the classical area, such as Naïve Bayesian method Cao (2010), but also in deep neural networks (VAE Kingma & Welling (2014), ZSL, etc.) over the last years. However, both have their shortages: Classical probability can not extract features from samples; For neural networks, the extracted features are usually abstract and cannot be directly used for numerical probability calculation. What if we combine them?

There are already some combinations of neural network and bayesian approach, such as probability distribution recognition Su & Chou (2006); Kocadağlı & Aşıkgil (2014), Bayesian approach are used to improve the accuracy of neural modeling Morales & Yu (2021), etc. However, current combinations do not take advantages of ZSL method.

We propose an approach to solve the mentioned problems, and we propose a novel unified combination of (indeterminate) probability theory and deep neural network. The neural network is used to extract attributes which are defined as discrete random variables, and the inference model for classification task is derived. Besides, these attributes do not need to be labeled in advance.

## E.2  RELATED WORK

**Tractable Probabilistic Models.**    There are a large family of tractable models including probabilistic circuits Choi et al. (2020); Dang et al. (2022), arithmetic circuits Darwiche (2002); Lowd & Domingos (2008), sum-product networks Poon & Domingos (2011), cutset networks Rahman et al. (2014), and-or search spaces Marinescu & Dechter (2005), and probabilistic sentential decision diagrams Kisa et al. (2014). The analytical solution of a probability calculation is defined as occurrence, $P(A = a) = \frac{\text{number of event } (A=a) \text{ occurs}}{\text{number of random experiments}}$, which is however not focused in these models. Our proposed IPNN is fully based on event occurrence and is an analytical solution.

**Deep Latent Variable Models.**    DLVMs are probabilistic models and can refer to the use of neural networks to perform latent variable inference Kim et al. (2018). Currently, the posterior calculation of continuous latent variables is regarded as intractable Kingma & Welling (2019), VAEs Kingma & Welling (2014); Titsias & Lázaro-Gredilla (2014); Rezende et al. (2014); Gregor et al. (2013) use variational inference method Jordan et al. (1999) as approximate solutions. Our proposed IPNN is one DLVM with discrete latent variables and the intractable posterior calculation is now analytically solved with our proposed theory.

### E.3 TRAINING

#### E.3.1 TRAINING STRATEGY

Given an input sample $x_t$ from a mini batch, with a minor modification of Equation (18):

$$P^{\mathbb{A}}\left(y_l \mid x_t\right) \approx \sum_{\mathbb{A}} \left( \frac{\max(H + h(\bar{t}), \epsilon)}{\max(G + g(\bar{t}), \epsilon)} \cdot \prod_{j=1}^{N} \alpha_{i_j}^j (t) \right) \tag{34}$$

$$h(\bar{t}) = \sum_{k=b \cdot (\bar{t}-1)+1}^{b \cdot \bar{t}} \left( y_l(k) \cdot \prod_{j=1}^{N} \alpha_{i_j}^j (k) \right) \tag{35}$$

$$g(\bar{t}) = \sum_{k=b \cdot (\bar{t}-1)+1}^{b \cdot \bar{t}} \left( \prod_{j=1}^{N} \alpha_{i_j}^j (k) \right) \tag{36}$$

$$H = \sum_{k=\max(1, \bar{t}-T)}^{\bar{t}-1} h(k), \text{for } \bar{t} = 2, 3, \dots \tag{37}$$

$$G = \sum_{k=\max(1, \bar{t}-T)}^{\bar{t}-1} g(k), \text{for } \bar{t} = 2, 3, \dots \tag{38}$$

Where $b$ is for batch size, $\bar{t} = \left\lceil \frac{t}{b} \right\rceil, t = 1, 2, \dots, n$. Hyperparameter T is for forgetting use, i.e., $H$ and $G$ are calculated from the recent T batches. Hyper-parameter T is introduced because at beginning of training phase the calculated result with Equation (10) is not good yet. And the $\epsilon$ on the denominator is to avoid dividing zero, the $\epsilon$ on the numerator is to have an initial value of 1. Besides, $H$ and $G$ are not needed for gradient updating during back-propagation. The detailed algorithm implementation is shown in Algorithm 1.

With Equation (34) we can get that $P^{\mathbb{A}}\left(y_l \mid x_1\right) = 1$ for the first input sample if $y_l$ is the ground truth and batch size is 1. Therefore, for IPNN the loss may increase at the beginning and fall back again while training.

---

**Algorithm 1** IPNN training

**Input**: A sample $x_t$ from mini-batch
**Parameter**: Split shape, forget number $T$, $\epsilon$, learning rate $\eta$.
**Output**: Posterior $P^{\mathbb{A}}\left(y_l \mid x_t\right)$

1: Declare default variables: $H, G, hList, gList$
2: **for** $\bar{t} = 1, 2, \dots$ Until Convergence **do**
3:  Compute $h, g$ with Equation (35) and Equation (36)
4:  Record: $hList.append(h), gList.append(g)$
5:  **if** $\bar{t} > T$ **then**
6:   Forget: $H = H - hList[0], G = G - gList[0]$
7:   Remove first element from $hList, gList$
8:  **end if**
9:  Compute posterior with Equation (34): $P^{\mathbb{A}}\left(y_l \mid x_t\right)$
10: Compute loss with Equation (19): $\mathcal{L}(\theta)$
11: Update model parameter: $\theta = \theta - \eta \nabla \mathcal{L}(\theta)$
12: Update for next loop: $H = H + h, G = G + g$
13: **end for**
14: **return** model and the posterior

---

#### E.3.2 MULTI-DEGREE CLASSIFICATION (OPTIONAL)

In IPNN, the model outputs N different random variables $A^1, A^2, \dots, A^N$, if we use part of them to form sub-joint sample spaces, we are able of doing sub classification task, the sub-joint spaces are defined as $\Lambda^1 \subset \mathbb{A}, \Lambda^2 \subset \mathbb{A}, \dots$ The number of sub-joint sample spaces is:

$$\sum_{j=1}^{N} \binom{N}{j} = \sum_{j=1}^{N} \left( \frac{N!}{j!(N-j)!} \right) \tag{39}$$

If the input samples are additionally labeled for part of sub-joint sample spaces[3], defined as $Y^\tau \in \{y_1^\tau, y_2^\tau, \dots, y_{m^\tau}^\tau\}$. The sub classification task can be represented as $\left\langle X, \Lambda^1, Y^1 \right\rangle, \left\langle X, \Lambda^2, Y^2 \right\rangle, \dots$ With Equation (19) we have,

$$\mathcal{L}^\tau = -\sum_{l=1}^{m^\tau} \left( y_l^\tau(k) \cdot \log P^{\Lambda^\tau}\left(y_l^\tau \mid x_t\right) \right), \tau = 1, 2, \dots \tag{40}$$

---

[3]It is labelling of input samples, not sub-joint sample points.

Together with the main loss, the overall loss is $\mathcal{L} + \mathcal{L}^1 + \mathcal{L}^2 + \dots$ In this way, we can perform multi-degree classification task. The additional labels can guide the convergence of the joint sample spaces and speed up the training process, as discussed later in Appendix E.7.1.

### E.3.3 MULTI-DEGREE UNSUPERVISED CLUSTERING

If there are no additional labels for the sub-joint sample spaces, the model are actually doing unsupervised clustering while training. And every sub-joint sample space describes one kind of clustering result, we have Equation (39) number of clustering situations in total.

### E.3.4 DESIGNATION OF JOINT SAMPLE SPACE

As in Appendix E.6 proved, we have following proposition:

**Proposition 2.** *For $P(y_l|x_k) = y_l(k) \in \{0,1\}$ hard label case, IPNN converges to global minimum only when $P\left(y_l|a_{i_1}^1, a_{i_2}^2, \ldots, a_{i_N}^N\right) = 1$, for $\prod_{j=1}^N \alpha_{i_j}^j(t) > 0$, $i_j = 1, 2, \ldots, M_j$. In other word, each joint sample point corresponds to an unique category. However, a category can correspond to one or more joint sample points.*

**Corollary 1.** *The necessary condition of achieving the global minimum is when the split shape defined in Equation (16) satisfies: $\prod_{j=1}^N M_j \geq m$, where $m$ is the number of classes. That is, for a classification task, the number of all joint sample points is greater than the classification classes.*

Theoretically, if model with 100 output nodes are split into 10 equal parts, it can classify 10 billion categories, validation result see Appendix E.7.1. Besides, the unsupervised clustering (Appendix E.3.3) depends on the input sample distributions, the split shape shall not violate from multi-degree clustering. For example, if the main attributes of one dataset shows three different colors, and your split shape is $\{2, 2, \dots\}$, this will hinder the unsupervised clustering, in this case, the shape of one random variable is better set to 3. And as in Appendix E.7 also analyzed, there are two local minimum situations, improper split shape will make IPNN go to local minimum.

In addition, the latter part from Proposition 2 also implies that IPNN may be able of doing further unsupervised classification task, this is beyond the scope of this discussion.

## E.4 RESULTS OF IPNN

### E.4.1 UNSUPERVISED CLUSTERING

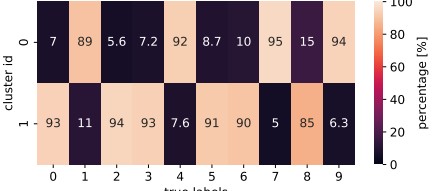
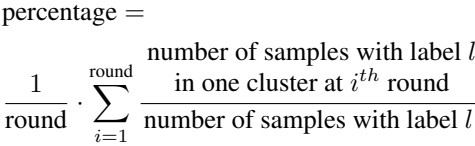

$$\text{percentage} =$$

$$\frac{1}{\text{round}} \cdot \sum_{i=1}^{\text{round}} \frac{\text{number of samples with label } l}{\text{number of samples with label } l}$$

Figure 4: Unsupervised clustering results on MNIST: test accuracy $95.1 \pm 0.4$, $\epsilon = 2$, batch size $b = 64$, forget number $T = 5$, epoch is 5 per round. The test was repeated for 876 rounds with same configuration (different random seeds) in order to check the stability of clustering performance, each round clustering result is aligned using Jaccard similarity Raff & Nicholas (2017).

As in Appendix E.3.3 discussed, IPNN is able of performing unsupervised clustering, we evaluate it on MNIST. The split shape is set to $\{2, 10\}$, it means we have two random variables, and the first random variable is used to divide MNIST labels $0, 1, \ldots 9$ into two clusters. The cluster results is shown in Figure 4.

We find only when $\epsilon$ in Equation (34) is set to a relative high value that IPNN prefers to put number 1,4,7,9 into one cluster and the rest into another cluster, otherwise, the clustering results is always different for each round training. The reason is unknown, our intuition is that high $\epsilon$ makes that each category catch the free joint sample point more harder, categories have similar attributes together will be more possible to catch the free joint sample point.

### E.4.2 Hyper-parameter Analysis

IPNN has two import hyper-parameters: split shape and forget number T. In this section, we have analyzed it with test on MNIST, batch size is set to 64, $\epsilon = 10^{-6}$. As shown in Figure 5a, if the number of joint sample points is smaller than 10, IPNN is not able of making a full classification and its test accuracy is proportional to number of joint sample points, as number of joint sample points increases over 10, IPNN goes to global minimum for both 3 cases, this result is consistent with our analysis. However, we have exceptions, the accuracy of split shape with $\{2, 5\}$ and $\{2, 6\}$ is not high. From Figure 4 we know that for the first random variable, IPNN sometimes tends to put number 1,4,7,9 into one cluster and the rest into another cluster, so this cluster result request that the split shape need to be set minimums to $\{2, \geq 6\}$ in order to have enough free joint sample points. That's why the accuracy of split shape with $\{2, 5\}$ is not high. (For $\{2, 6\}$ case, only three numbers are in one cluster.)

Another test in Figure 5b shows that IPNN will go to local minimum as forget number T increases and cannot go to global minimum without further actions, hence, a relative small forget number T shall be found with try and error.

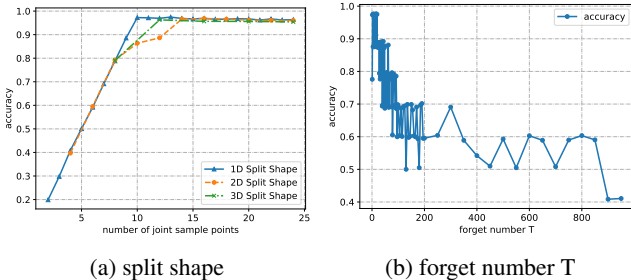

|(a) split shape | (b) forget number T|

Figure 5: (a) Impact Analysis of split shape with MNIST: 1D split shape is for $\{\tau\}, \tau = 2, 3, \ldots, 24$. 2D split shape is for $\{2, \tau\}, \tau = 2, 3, \ldots, 12$. 3D split shape is for $\{2, 2, \tau\}, \tau = 2, 3, \ldots, 6$. The x-axis is the number of joint sample points calculated with $\prod_{j=1}^{N} M_j$, see Equation (16).
(b) Impact Analysis of forget number T with MNIST: Split shape is $\{10\}$.

### E.5 Conclusion

For a classification task, we proposed an approach to extract the attributes of input samples as random variables, and these variables are used to form a large joint sample space. After IPNN converges to global minimum, each joint sample point will correspond to an unique category, as discussed in Proposition 2. As the joint sample space increases exponentially, the classification capability of IPNN will increase accordingly.

We can then use the advantages of classical probability theory, for example, for very large joint sample space, we can use the Bayesian network approach or mutual independence among variables (see Appendix E.8) to simplify the model and improve the inference efficiency, in this way, a more complex Bayesian network could be built for more complex reasoning task.

### E.6 Global Minimum Analysis

*Proof of Proposition 2.* Equation (18) can be rewritten as:

$$P^{\mathbb{A}}(y_l \mid x_t) = \sum_{\mathbb{A}} \left( p_{\mathbb{A}} \cdot \prod_{j=1}^{N} \alpha_{i_j}^{j}(t) \right) \tag{41}$$

Where,

$$p_{\mathbb{A}} = P\left(y_l \mid a_{i_1}^1, a_{i_2}^2, \ldots, a_{i_N}^N\right) \tag{42}$$

Theoretically, for $P(y_l|x_k) = y_l(k) \in \{0, 1\}$ hard label case, model converges to global minimum when the train and test loss is zero Li & Yuan (2017), and for the ground truth $y_l(t) = 1$, with Equation (19) we have:

$$\sum_{\mathbb{A}} \left( p_{\mathbb{A}} \cdot \prod_{j=1}^{N} \alpha_{i_j}^{j}(t) \right) = 1 \tag{43}$$

Subtract the above equation from Equation (6) gives:

$$\sum_{\mathbb{A}} \left( (1 - p_{\mathbb{A}}) \cdot \prod_{j=1}^{N} \alpha_{i_j}^{j}(t) \right) = 0 \tag{44}$$

Because $\prod_{j=1}^{N} \alpha_{i_j}^{j}(t) \in [0, 1]$ and $(1 - p_{\mathbb{A}}) \in [0, 1]$, The above equation is then equivalent to:

$$p_{\mathbb{A}} = 1, \text{ for } \prod_{j=1}^{N} \alpha_{i_j}^{j}(t) > 0, i_j = 1, 2, \ldots, M_j. \tag{45}$$

$\square$

### E.7 LOCAL MINIMUM ANALYSIS

Equation (41) can be further rewritten as:

$$P^{\mathbb{A}}(y_l \mid x_t) = \sum_{i_\tau=1}^{M_\tau} \left( \alpha_{i_\tau}^{\tau}(t) \cdot \sum_{\Lambda} \left( p_{\mathbb{A}} \cdot \prod_{j=1, j\neq\tau}^{N} \alpha_{i_j}^{j}(t) \right) \right) = \sum_{i_\tau=1}^{M_\tau} \left( \alpha_{i_\tau}^{\tau}(t) \cdot p_{i_\tau} \right) \tag{46}$$

Where $\Lambda = (A^1, \ldots, A^j, \ldots, A^N) \subset \mathbb{A}, j \neq \tau$ and,

$$p_{i_\tau} = \sum_{\Lambda} \left( p_{\mathbb{A}} \cdot \prod_{j=1, j\neq\tau}^{N} \alpha_{i_j}^{j}(t) \right) \tag{47}$$

Substitute Equation (46) into Equation (19), and for the ground truth $y_l(t) = 1$ the loss function can be written as:

$$\mathcal{L} = -\log\left( \sum_{i_\tau=1}^{M_\tau} \left( \alpha_{i_\tau}^{\tau}(t) \cdot p_{i_\tau} \right) \right) \tag{48}$$

Let the model output before softmax function be $z_{i_j}$, we have:

$$\alpha_{i_\tau}^{\tau}(t) = \frac{e^{z_{i_\tau}}}{\sum_{i_j=1}^{M_j} e^{z_{i_j}}} \tag{49}$$

In order to simplify the calculation, we assume $p_{\mathbb{A}}$ defined in Equation (42) is constant during back-propagation. so the gradient is:

$$\frac{\partial \mathcal{L}}{\partial z_{i_\tau}} = -\frac{\alpha_{i_\tau}^{\tau}(t) \cdot \sum_{i_j=1, i_j\neq i_\tau}^{M_j} \left( e^{z_{i_j}} \cdot (p_{i_\tau} - p_{i_j}) \right)}{\sum_{i_j=1}^{M_j} \left( e^{z_{i_j}} \cdot p_{i_j} \right)} \tag{50}$$

Therefore, we have two kind of situations that the algorithm will go to local minimum:

$$\frac{\partial \mathcal{L}}{\partial z_{i_\tau}} = \begin{cases} \to 0, & \text{if } \left| z_{i_\tau} - z_{i_j} \right| \to \infty \\ 0, & \text{if } p_{i_\tau} = p_{i_j} \\ Nonezero, & o.w. \end{cases} \tag{51}$$

Where $i_\tau = 1, 2, \ldots, M_\tau$.

The first local minimum usually happens when Corollary 1 is not satisfied, that is, the number of joint sample points is smaller than the classification classes, the results are shown in Figure 5a.

If the model weights are initialized to a very small value, the second local minimum may happen at the beginning of training. In such case, all the model output values are also small which will result in $\alpha_1^j(t) \approx \alpha_2^j(t) \approx \cdots \approx \alpha_{M_j}^j(t)$, and it will further lead to all the $p_{i_\tau}$ be similar among each other. Therefore, if the model loss reduces slowly at the beginning of training, the model weights is suggested to be initialized to an relative high value. But the model weights shall not be set to too high values, otherwise it will lead to first local minimum.

As shown in Figure 6, if model weights are initialized to uniform distribution of $\left[-10^{-6}, 10^{-6}\right]$, its convergence speed is slower than the model weights initialized to uniform distribution of $[-0.3, 0.3]$. Besides, model weights initialized to uniform distribution of $[-3, 3]$ get almost stuck at local minimum and cannot go to global minimum. This result is consistent with our analysis.

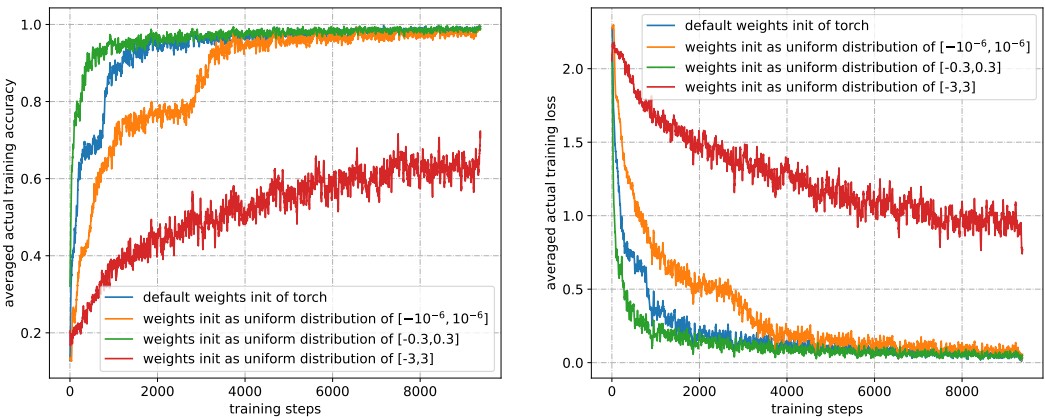

Figure 6: Model weights initialization impact analysis on MNIST. Split shape is $\{2, 10\}$, batch size is 64, forget number $T = 5$, $\epsilon = 10^{-6}$.

### E.7.1 Avoiding Local Minimum with Multi-degree Classification

Another experiment is designed by us to check the performance of multi-degree classification (see Appendix E.3.2): classification of binary vector into decimal value. The binary vector is the model inputs from '000000000000' to '111111111111', which are labeled from 0 to 4095. The split shape is set to $\{M_1 = 2, M_2 = 2, \ldots, M_{12} = 2\}$, which is exactly able of making a full classification. Besides, model weights are initialized as uniform distribution of $[-0.3, 0.3]$, as discussed in Appendix E.7.

The result is shown in Figure 7, IPNN without multi degree classification goes to local minimum with only $69.5\%$ train accuracy. We have only additionally labeled for 12 sub-joint spaces, and IPNN goes to global minimum with $100\%$ train accuracy.

Therefore, with only $\sum_1^{12} 2 = 24$ output nodes, IPNN can classify 4096 categories. Theoretically, if model with 100 output nodes are split into 10 equal parts, it can classify 10 billion categories. Hence, compared with the classification model with only one 'softmax' function, IPNN has no computationally expensive problems (see Section 1).

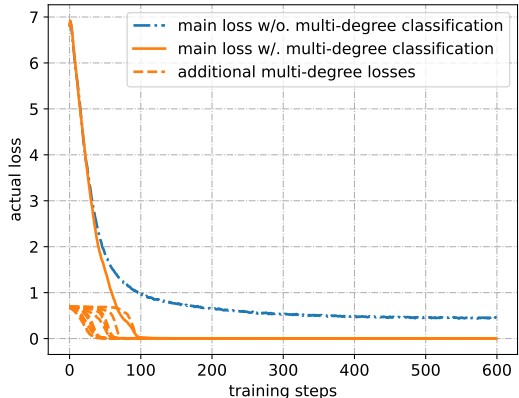

Figure 7: Loss of multi-degree classification of 'binary to decimal' on train dataset. Input samples are additionally labeled with $Y^i \in \{0, 1\}$ for $i^{th}$ bit is 0 or 1, respectively. $Y^i$ corresponds to sub-joint sample space $\Lambda^i$ with split shape $\{M_i = 2\}, i = 1, 2, \ldots 12$. Batch size is 4096, forget number $T = 5, \ \epsilon = 10^{-6}$.

### E.8 MUTUAL INDEPENDENCY

If we want the random variables $A^1, A^2, \ldots, A^N$ partly or fully mutually independent, we can use their mutual information as loss function:

$$
\mathcal{L}^* = KL\left(P(A^1, A^2, \ldots, A^N), \prod_{j=1}^{N} P(A^j)\right) = \sum_{\mathbb{A}} \left(P\left(a_{i_1}^1, \ldots, a_{i_N}^N\right) \cdot \log \frac{P\left(a_{i_1}^1, \ldots, a_{i_N}^N\right)}{\prod_{j=1}^{N} P(a_{i_j}^j)}\right)
$$

(52)

$$
= \sum_{\mathbb{A}} \left(\frac{\sum_{k=1}^{n}\left(\prod_{j=1}^{N} \alpha_{i_j}^j(k)\right)}{n} \cdot \log\left(\frac{\frac{\sum_{k=1}^{n}\left(\prod_{j=1}^{N} \alpha_{i_j}^j(k)\right)}{n}}{\prod_{j=1}^{N} \frac{\sum_{k=1}^{n} \alpha_{i_j}^j(k)}{n}}\right)\right)
$$

### E.9 LIMITATIONS

**Indeterminate Probability Theory.** As we summarized in Section 3.5, we do not find any exceptions for our proposed three conditional mutual independency assumptions, see Candidate Axiom 1 Candidate Axiom 2 and Candidate Axiom 3. And our proposed Equation (15) is derived from these assumptions, in our opinion, this equation can be applied to any general random experiment.

**IPNN.** IPNN is one neural network framework based on indeterminate probability theory, it has three limitations: (1) The split shape need to be predefined, a proper sample space for an unknown dataset can only be found with try and error. The latent variables are continuous in CIPNN Anonymous (2024a), therefore this issue does not exist in CIPNN. (2) It sometimes converges to local minimum, but we can avoid this problem with a proper model weights initialization, as discussed in Appendix E.7. (3) As joint sample space increases exponentially, the memory consumption and computation time also increase accordingly. This issue only exist during training, and can be avoided through monte carlo method for prediction task, as discussed in CIPNN Anonymous (2024a), this paper will not further discuss it.

### E.10 PSEUDO CODE PYTORCH IMPLEMENTATION OF IPNN

```
'''
Pseudo code of calculation of the loss and the inference posterior
    P^{A}(Y|X).
```

```
b                        --> batch size
y                        --> number of classification classes
[M_1, M_2, ..., M_N] --> split shape

inputs:
   logits: [b, M_1 + M_2 +, ..., M_N] # neural network outputs
   y_true: [b,y] # labels
outputs:
   probability: [b,y] # the inference posterior P^{A}(Y|X)
   loss
'''

logits = torch.split(logits, split_shape, dim = -1)
# Shape of variables: [[b, M_1], [b, M_2], ..., [b, M_N]]
variables = [torch.softmax(_,dim = -1) for _ in logits]

# Joint sample space calculation
# Shape of joint_variables: [b, M_1, M_2, ..., M_N]
for i in range(len(variables)):
if i == 0 :
     joint_variables = variables[i]
else:
     r_ = EINSUM_CHAR[:joint_variables.dim()-1]
     joint_variables = torch.einsum('b{},ba->b{}a'.format(r_,r_),
         joint_variables,variables[i]) # see Equation (5)

# OBSERVATION PHASE
r_ = EINSUM_CHAR[:joint_variables.dim()-1]
num_y_joint_current = torch.einsum('b{},by->y{}'.format(r_,r_),
   joint_variables,y_true) #  # see Equation (35)
num_joint_current = torch.sum(joint_variables,dim = 0) # see
   Equation (36)

# numerator and denominator of conditional probability P(Y|A^1,A
   ^2,...,A^N)
num_y_joint += num_y_joint_current # see Equation (37)
num_joint += num_joint_current # see Equation (38)

# Shape of prob_y_joint: [y, M_1, M_2, ..., M_N]
prob_y_joint = num_y_joint / num_joint # see Equation (10)

# INFERENCE PHASE
# Shape of probability: [b,y]
r_ = EINSUM_CHAR[:joint_variables.dim()-1]
probability = torch.einsum('y{},b{}->by'.format(r_,r_),
   prob_y_joint,joint_variables) # see Equation (34)

# loss function
loss = cross_entropy_loss(probability,y_true) # see Equation (19)
```

