# OpenReview forum: "Indeterminate Probability Theory"
_ICLR.cc/2024/Conference — Submitted to ICLR 2024_

### Official Review · Reviewer_S3ce · 2023-10-18

**Soundness:** 2 fair
**Presentation:** 1 poor
**Contribution:** 1 poor
**Rating:** 3
**Confidence:** 3

**Summary:**

This paper proposes "Indeterminate Probability Theory", which is claimed as an extension of classical probability theory. Based on the proposed theory, the authors derive an analytical expression of general posterior, which has some applications such as IPNN and CIPNN. Experimental results validate the proposed theory.

**Strengths:**

This paper claims to extend classical probability theory, which is very ambitious and is definitely important if this is true.

**Weaknesses:**

The main contribution of this paper is the proposed "Indeterministic Probability Theory", but it is far from satisfaction to be a theory, especially when it is stated to be "an extension of classical probability theory". It is actually built on the axioms of classical probability theory, added with a specific generation process of random variables, and three proposed "candidate axioms", thus it at most becomes "a special sub-field of classical probability theory".

Even worse, the paper claims that "our most important contribution is that we propose a new **general analytical** and **tractable** probability equation", but neither is theoretically validated: for **general analytical**, it is analytical, but it is not discussed enough why the proposed two-phase protocol is general; for **tractable**, it is also not verified the error of approximation via Monte Carlo methods. There are some experimental results to verify the effectiveness of Monte Carlo, but is over-simplified to validate it in such a general theory as is claimed, and more importantly, the effectiveness of Monte Carlo in this paper is not "proved" yet.

To be honest, section 3 is more like a section of "problem formulation + proposed approach": The two-phase protocol is more like the problem formulation, the axioms are more like some assumptions of independence, and the complexity reduction using Monte Carlo is more like the proposed approach.

**Questions:**

What do you want to say in Section 2? It seems that the example does not go beyond classical probability theory, i.e., all definitions, quantities and calculations are consistent with definitions and axioms in classical probability theory.

What is new in your indeterminate probability theory? Specifically, I am confused why eq. (4) must be 0 or 1 in classical probability theory. Could the authors give some references? The authors should give references, clear derivations or rigorous counter-examples when refuting something in classical probability theory, as it is based on rigorous mathematics.

How general your proposed theory is? For example, does your theory enable A and Y to be any kind of random variables, and can the two-phase protocol in your theory model any data-generation process? If not, then the generality of your theory should be discussed.

In page 5 the authors say "...Otherwise, Candidate Axiom 2 and Candidate Axiom 3 cannot both be true". In my opinion, it is strange to discuss the soundness of an axiom once it is proposed.

---

> ### Author Response · Authors · 2023-11-17
> **Response to Reviewer S3ce**
>
> Dear Reviewer S3ce,
>
> Thank you very much for your detailed feedbacks.
>
> **Q1**: What do you want to say in Section 2? It seems that the example does not go beyond classical probability theory, i.e., all definitions, quantities and calculations are consistent with definitions and axioms in classical probability theory.
>
> A1: The example in Section 2 employs our proposed candidate axiom 3, which serves as a helpful introduction to our theory. Our intention is to provide readers with a seamless transition into our theory by using this example.
>
> **Q2**: What is new in your indeterminate probability theory?
>
> A2: We suggest that you take a look at the simple coin toss example in Appendix B. If you can solve it using other probability theories, then our theory would not offer anything new.
>
> **Q3**: Specifically, I am confused why eq. (4) must be 0 or 1 in classical probability theory. Could the authors give some references?
>
> A3: The point here is that Eq. (1) in the paper is only applicable when Eq. (4) is 0 or 1.
>
> **Q4**: The authors should give references, clear derivations or rigorous counter-examples when refuting something in classical probability theory, as it is based on rigorous mathematics.
>
> A4: Our proposed equation does not refute anything in classical probability theory. The classical probability Eq. (1) is a special case to our equation.
>
>
> **Q5**: How general your proposed theory is? For example, does your theory enable A and Y to be any kind of random variables, and can the two-phase protocol in your theory model any data-generation process?
>
> A5: The generalization of our theory is a conclusion from our proposed new candidate axioms. Therefore, if even a single exception is found, it would not be appropriate to assert that the theory is general.
> Moreover, we have applied our theory to thousand dimensional continuous latent spaces without any special requirements for the variables or datasets. That is why we say that the proposed equation is general.
>
> **Q6**: In page 5 the authors say "...Otherwise, Candidate Axiom 2 and Candidate Axiom 3 cannot both be true". In my opinion, it is strange to discuss the soundness of an axiom once it is proposed.
>
> A6: We have a differing opinion on this point. Since the axioms are newly proposed, readers should initially approach them with skepticism. The soundness of new axioms should not only be discussed, but also given the highest priority for verification, as they serve as the foundation for a new theory. (For example, it is wrong to claim naïve Bayes independent assumption as an axiom.)
>
> Best regards
>
> Authors

---

> > ### Comment · Reviewer_S3ce · 2023-11-17
> >
> > I appreciate the authors for the detailed response. I am writing to let the authors know that I have noticed the authors rebuttal. I promise to read the paper again with the help of authors' response, and will reply in one or two days, as soon as possible. Thanks again!

---

> > ### Comment · Reviewer_S3ce · 2023-11-18
> >
> > I agree with the authors on the answer to my quesion Q1 and Q6. For the rest, please refer to the following:
> >
> > To be very honest, I still cannot get that what is really new in the proposed "new theory", especially when the authors place it with the celebrated field of probability theory.
> >
> > **About probability theory.** I would first apologize for my incorrect wording "classical probability theory": I'm actually referring to the field of probability theory based on measure theory (advanced probability theory, in some courses). Thus, I am very surprised when the authors claimed to have established a theory in several pages without taking limits and integrations, and even make the existing probability theory becoming its special case. I still do not find any expressions in the paper that cannot be written in existing definitions of probability theory, so I keep my concern on whether it is a new probability theory. There are massive textbooks and lessons on "advanced probability theory", so I would not list the definitions of it. The authors could also kindly refer to the first reply from Reviewer Xi4x, especially the first and the fourth points.
> >
> > **About the contribution of this paper.** Given that this paper actually does not extend the existing probability theory (I am not sure on this now), it would be more accepting if the main contribution is a new approach to estimate general posterior. If so, the authors must clearly discuss related works.
> >
> > Thanks!

---

> > > ### Author Response · Authors · 2023-11-20
> > > **2nd Response to Reviewer S3ce**
> > >
> > > Dear Reviewer S3ce,
> > >
> > > Thank you very much for your feedback and the additional effort you made to reread our paper. We are very grateful for your efforts, especially for using your weekend time.
> > >
> > > **2nd_Q1**: about measure theory.
> > >
> > > 2nd_A1: We received the same suggestion to formulate our theory according to measure theory in our last submission, and we responded that we do not know how to formulate it using that approach. As the first author of this paper, my knowledge of it is limited. I believe that even if I were to formulate the theory using measure theory, the writing would still likely be regarded as 'not rigorous'. At least with the current formulation, I am able to follow it well.
> > >
> > > **2nd_Q2**: The authors could also kindly refer to the first reply from Reviewer Xi4x, especially the first and the fourth points.
> > > 2nd_A2: Please refer to the answer to Reviewer Xi4x.
> > >
> > > Best regards
> > >
> > > Authors

---

> > > > ### Comment · Reviewer_S3ce · 2023-11-21
> > > >
> > > > Thanks for your response.

---

### Official Review · Reviewer_Xi4x · 2023-10-29

**Soundness:** 2 fair
**Presentation:** 1 poor
**Contribution:** 1 poor
**Rating:** 1
**Confidence:** 5

**Summary:**

The paper proposes to introduce a new theory of probability to cope with imperfect observations, in the sense that the reported value can be different from the true experimental result. This is done through the introduction of an "observer", which can be imperfect, in the sense that it is noisy. The theory is then applied to various case studies.

**Strengths:**

I do not really perceive any strong point in the paper, other than the fact that modelling imperfect observational process is an interesting, yet arguably old topic.

**Weaknesses:**

This paper is puzzling me in more than one ways, and I will focus on the main ones (some for which the authors can offer a rebuttal, mostly when it concerns the content and not the form of the paper).

A first thing is that the paper is written in a very unusual way, at least for a paper of computer science and/or machine learning. It is very rare to directly start with a mathematical formulation, without making first an introduction (and possibly related work) positioning the proposal and its originality.

A second thing is that the paper is very quick on some technical details, while being very verbose on rather basing thing such as classical applications of probabilistic conditioning. It is also a bit cryptic in terms of language as well as bit naive about some aspects. For instance, P2 top, it is not true that one cannot apply Bayes rule in continuous setting, and it has been numerous, numerous times. At this point, what means indeterminate is also quite obscure. Similarly, it is not clear for the naive Bayes what exactly means $P(A^j=a^j_{i_j}|Y=y_l)$ being not solvable? It can certainly be estimated from data, even in case of noisy observations or untrue assumptions (potentially leading to biased estimates, but it can nonetheless be estimated).

A third thing is that it is unclear what authors really understand by “indeterminate”: is it that the observational process is noisy, or that the obtained probabilities are ill-known and hence that one should consider sets of possible probabilities? The paper suggests the first case, yet in such a situation I really do not see what is different between what is proposed in the paper and the consideration of noisy data where one does know or can estimate the noise process? Given that there is a huge literature on learning from noisy (and/or imprecise) data, at least a positioning with respect to those should be done. Indeed, if the main idea of the paper is to have $P(y_{obs}=y|y_{true}=y)<1$ ($y$ here can be either the output value or a feature value) and then to proceed from that, then I would argue that considering such a situation is not new at all. Similarly, if indeterminate means ill-defined, then there is a whole literature about that (see, e.g., work following the book on Peter Walley on imprecise probabilities and similar). Claiming to build a new theory of probability should be backed up by being very precise about why previous theories do not answer the considered problem.

A fourth thing is that it is really unclear to me why the current experiments, that merely show accuracy results for standard problems, do show that the theory is “valid”? I would equally question a statistical learning theory or more generally an uncertainty theory whose axioms cannot be the subject of tests and falsification? All theories of uncertainty I know of that are a bit serious in terms of operationally are subject to falsifiability, and this especially true for probabilistic theories (see the Ellsberg paradox for a good example of attempted falsification). Also, since Softmax does not enjoy peculiarly good properties from a theoretical perspective, I would not consider it as a strong baselines against which to test the axioms of a theory?

**Questions:**

See weaknesses

**Details Of Ethics Concerns:**

No ethical concerns

---

> ### Author Response · Authors · 2023-11-17
> **Response to Reviewer Xi4x**
>
> Dear Reviewer Xi4x,
>
> Thank you very much for your detailed feedbacks.
>
> **Q1**: For instance, P2 top, it is not true that one cannot apply Bayes rule in continuous setting, and it has been numerous, numerous times.
>
> A1: We respectfully wish to remind you that you have misunderstood our point in P2 top. We merely stated that continuous variable is not applicable to Eq. (1), not Bayes rule.
>
> **Q2**: it is not clear for the naive Bayes what exactly means $P(A^j=a^j_{i_j}|Y=y_l)$ being not always solvable?
>
> A2: The coin toss example in Appendix B is an evidence to this point.
>
> **Q3**: is it that the observational process is noisy, or that the obtained probabilities are ill-known and hence that one should consider sets of possible probabilities?
> ...
> if the main idea of the paper is to have $P(y_{obs}=y|y_{true}=y)<1$ ($y$ here can be either the output value or a feature value)?
>
> A3: Your formulation does not align with our theory. Hence, we will adopt your idea to frame our case as follows: $P(y_{obs}=y|x_{k})<1$ and $P(y_{true}=y|x_{k})=1$. Our focus is solely on the indeterminate observation for the $k^{th}$ random experiment. And our objective is to utilize $A_{obs}$ ($P(A_{obs}=a|x_{k})<1$) to infer $y_{obs}$. More details you can find in Q5 of Reviewer MWjN.
>
>
> **Q4**: Similarly, if indeterminate means ill-defined, then there is a whole literature about that (see, e.g., work following the book on Peter Walley on imprecise probabilities and similar).
>
> A4: Thank you for sharing Peter Walley's imprecise probabilities theory. From our limited understanding, this theory differs from ours in that it represents probabilities using sets or intervals of values, rather than a single point.
>
> **Q5**: I would equally question a statistical learning theory or more generally an uncertainty theory whose axioms cannot be the subject of tests and falsification?
>
> A5: To deny a new axiom, all you need is to find a single exception, even a simple toy example would suffice.
>
> **Q6**: All theories of uncertainty I know of that are a bit serious in terms of operationally are subject to falsifiability, and this especially true for probabilistic theories (see the Ellsberg paradox for a good example of attempted falsification).
>
> A6: Thank you for sharing the Ellsberg paradox with us. However, this example differs from our case. The reason why Axioms 1, 2, and 3 cannot be falsified is that they have no exceptions.
>
> **Q7**: since Softmax does not enjoy peculiarly good properties from a theoretical perspective,
>
> A7: We agree that softmax properties are not good enough. In our applications, softmax is only used for IPNN, and we have encountered local minimum problems with this function, as outlined in Appendix E.7. However, our CIPNN and MTS forecasting method rely on Gaussian distribution, not softmax function.
>
>
> Best regards
>
> Authors

---

> ### Comment · Reviewer_Xi4x · 2023-11-17
> **A first reply to clarify some points**
>
> Dear authors,
>
> Thanks for your various replies to my questions. Before examining them and the paper again (two things I will only be able to do with the limited time I can allocate to discuss/review a specific ICLR paper), I would like to point out that my initial comments aimed at pointing out general problems of the paper, and what authors have provided are rather specific answers compared to this generality. My current opinion is that before being accepted, the paper would need to clarify many things:
>
> * First, being crystal clear about the mathematical statements made in the paper. This is especially important for axiomatic and theoretic studies. Such a thing as "Not applicable if $A_j$ is continuous" will not really make it, especially as frequentist interpretation of probabilities (one possible interpretation of Equation (1)) can perfectly deal with continuous spaces.
>
> * Second, and as the work intends to be axiomatic, be very clear about the terminology. As said in the review, the word "indeterminate" usually refers to coarse/missing/imprecise aspects, something that is not really discussed here. So there is a need at least to explain how the notion of indeterminacy here differs from the classical notions of indeterminacy.
>
> * Third positioning itself from other works that are very close to the presented idea, such as noisy observations (note that noise can be defined instance-wise as well).
>
> * Fourth, all of this (clear positioning, unanmbiguous mathematics for all critics/statements) are necessary elements of a paper advancing a new theory/framework, that should not be relegated to the appendices for the sake of space limitations. As authors indicate by referring very often to the appendix, this may be something that is not doable in a conference format, and rather than potentially making a fourth submission to any A/A* conference of their preference, maybe a full journal paper (or a book, if they really want to expose a new, full-fledged theory) would be more suitable. On one side it would at least give more time to reviewers to fully read the paper and enter into the examples, and on the other side it would allow the authors to make a full, complete exposure of their work.
>
> * Fifth and not last, I definitely remain wary of an uncertainty theory where "Axioms 1, 2, and 3 cannot be falsified is that they have no exception", at least for two reasons: this would suggest a "perfect theory", which arguably cannot exists, and also as I've said, I think a theory about uncertainty should be falsifiable to some extent, so as to know its limits and be open to criticism.
>
> To be clear, I am not advancing that there is no value in the current paper (there may well be), but if there is, the current presentation does not allow me to assess them clearly (in particular in comparison to the very, very vast literature on probability foundations).
>
> Best

---

> > ### Author Response · Authors · 2023-11-20
> > **2nd Response to Reviewer Xi4x**
> >
> > Dear Reviewer Xi4x,
> >
> > Thank you very much for your quick and detailed feedback, we are sorry for our late response.
> >
> > As the first author of this paper, the new equation is found occasionally when I design new mode architecture.
> > To be honest, I don't have a strong background in the field of probability and I agree that the writing problems are a shortage to this paper.
> >
> > **2nd_Q1**:  ... frequentist interpretation of probabilities (one possible interpretation of Equation (1)) can perfectly deal with continuous spaces.
> >
> > 2nd_A1: As I understood, Eq. (1) is primarily designed for discrete random variables (It seems that I made a mistake here, I will change the formulation to this point later.). And I agree that the mathematical statements are a problem.
> >
> > **2nd_Q2**: As said in the review, the word "indeterminate" usually refers to coarse/missing/imprecise aspects, something that is not really discussed here.
> >
> > 2nd_A2: I did not respond to your general questions in our initial response, because I don't know how to offer a rebuttal to this point. (As you have written 'some for which the authors can offer a rebuttal, mostly when it concerns the content and *not the form of the paper*')
> >
> > **2nd_Q3**: Third positioning itself from other works that are very close to the presented idea, such as noisy observations (note that noise can be defined instance-wise as well).
> >
> > 2nd_A3: To my understanding, the 'indeterminate' is only a similar word to 'noisy', it is also difficult for me to connect this paper to it. For example, the outcome of a Coin toss example in this paper is interpreted as a Gaussian distribution, this is hard to make a connection to noisy observations.
> >
> > **2nd_Q4**: ... that should not be relegated to the appendices for the sake of space limitations.
> >
> > 2nd_A4: I apologize for these references to appendix.
> >
> > **2nd_Q5**: "Axioms 1, 2, and 3 cannot be falsified is that they have no exception"...so as to know its limits and be open to criticism.
> >
> > 2nd_A5: In my opinion, this is the most important contributions of our paper. If the proposed candidate axioms have exception and can be falsified, it would be not correct to refer to them as axioms.
> > Besides, I am open to criticism, I write these independent assumptions as **candidate** axioms, because I think they are very important. And these candidate axioms will remain valid until someone finds an exception.
> >
> > Finally, as you have written, if these writing or formulation problems are considered as a critical reason for rejection, I don't think I can offer a rebuttal for this point.
> >
> > Best regards
> >
> > Authors

---

### Official Review · Reviewer_MWjN · 2023-10-31

**Soundness:** 3 good
**Presentation:** 3 good
**Contribution:** 2 fair
**Rating:** 6
**Confidence:** 4

**Summary:**

This is an extremely ambitious paper that attempts to construct a new theory called indeterminate probability theory. The key idea of Indeterminate probability theory is to introduce a new concept of auxiliary observers and to treat the results of each random experiment as an indeterminate probability distribution, while still preserving the assumption of mutual independence. As a result, the posterior probabilities of the system can be derived in a form that is easy to handle analytically, an important benefit in applications.
The authors demonstrate the applicability of this idea to regression and classification problems by combining it with neural networks.

**Strengths:**

I am very grateful to the authors for sharing their novel attempt at this paper. I enjoyed reading this paper very much.
- The paper devotes a great deal of effort in its presentation to illustrate new ideas that are outside of the conventional wisdom. The paper is very well written and its organization is designed to appeal to a diverse audience. In particular, it is designed to be easily understood by explaining the core ideas by means of toy examples.
- The practical contribution of this paper is very significant. Traditionally, posterior probabilities in statistical machine learning have been approximated by some kind of approximation method (e.g., Markov chain Monte Carlo or variational methods), but the ideas in this paper have the potential to be a new option to add to that.

**Weaknesses:**

First of all, let me emphasize that I am trying to be very open minded in understanding the value of this paper. My grade on my first peer review may not be very high, but I am prepared to improve it as soon as I properly understand the value of this paper.
My concern is whether this paper could create a new system of probability theory (i.e., a major historical breakthrough) or whether it provides a new perspective on approximation and interpretation for the system in a form that is easy to handle in applications (i.e., a new alternative alongside MCMC and VB), a somewhat excessive Is it an appealing proposition? I would like to inquire in the question section for more details.

**Questions:**

My question can be summarized very simply as to whether or not indeterminate probability theory can be expressed in terms of a definition of probability space using abstract probability space.

First of all, I understand this new insightful strategy of the authors as follows (Perhaps this understanding of mine is incorrect. If I am wrong, I would be very grateful if you could correct me.)
- The authors' system introduces uncertainty as an auxiliary variable for observers. If this were to be expressed in the context of a conventional standard Bayesian analysis, the observer could be represented as making an observation error according to the auxiliary random variable.
- Next, since this auxiliary random variable is not needed to describe the system, we will try to eliminate it in some way. In a conventional standard Bayesian analysis, this can be done by eliminating the auxiliary random variable by marginalization. However, a problem arises here. If the auxiliary random variable is shared by all observers, the system loses observer independence (Axiom 2 of the proposed probability theory) when it is eliminated.
- Therefore, the proposed probability theory simply ignores the auxiliary random variable while simultaneously assuming Axiom 2.

If we were to use such a strategy, it would certainly seem that we could view the system as different from classical probability theory (as mentioned in the paper, we could of course make special cases that are equivalent to classical probability theory in special circumstances).

Following this intuition, my interest is in what the authors' system would look like if it were represented in an abstract probability space. That is, a situation where all randomness in the world is governed by an abstract space $\Theta$, where all randomness is lost if the abstract space is determined at a point $\theta\in\Theta$, and where all variables can be described deterministically. In the abstract space, random variables are represented as a projection of the world as a map to an object, e.g., $Y(\theta), X(\theta), A(\theta)$ can be uniquely determined for a given source $\theta$. Can the authors' system be represented using such a conventional abstract probability space? Or is it a deviation from that rule?

---

> ### Author Response · Authors · 2023-11-17
> **Response to Reviewer MWjN**
>
> Dear Reviewer MWjN,
>
> We sincerely appreciate your insightful comments, and we are excited to receive such in-depth feedbacks.
> We appreciate your high-level discussion on this topic, as well as the new perspective in which you summarized the theory in the 'Questions' part.
> Here are our detailed responses to your questions.
>
>
> **Q1**: ... if it can be expressed as abstract probability space.
>
> A1: Yes, we will provide a detailed response starting from Q3.
>
> **Q2**: your own understanding ... the proposed probability theory simply ignores the auxiliary random variable while simultaneously assuming Axiom 2.
>
> A2: We are surprised by your new understanding of this theory, and we agree with you.
> We will provide further details to support this understanding.
>
> $$
> \underset{\text{(a)}}{\underbrace{{\color{Red}P^{\mathbb{A}}\left ( y_{l}\mid x_{n+1}   \right ) }}}
> = \underset{\text{(b)}}{\underbrace{\sum_{\mathbb{A}}P\left ( y_{l},\mathbb{A}\mid x_{n+1}   \right )}}
> =\underset{\text{(c)}}{\underbrace{
>     \sum_{\mathbb{A}}\left (P\left ( y_{l}\mid \mathbb{A} \right )
> \cdot P(\mathbb{A}\mid x_{n+1}) \right )}}
> =\underset{\text{(d)}}{\underbrace{
>     \sum_{\mathbb{A}}\left (\frac{ {\textstyle \sum_{k=1}^{n}\left (
>     {\color{Red} P(y_{l}\mid x_{k})}\cdot  P(\mathbb{A}\mid x_{k}) \right ) } } {{\textstyle \sum_{k=1}^{n}P(\mathbb{A}\mid x_{k})  } }
> \cdot P(\mathbb{A}\mid x_{n+1}) \right )}}
> = \underset{\text{(e)}}{\underbrace{\text{Eq. (15)}}}
> $$
>
> From (a) to (b): marginalization.
> From (b) to (c): candidate axiom 3.
> From (c) to (d): candidate axiom 2.
> From (d) to (e): candidate axiom 1.
>
> Where superscript $\mathbb{A}$ serves only as an indicator to distinguish the two red terms. For continuous random variables, we change $\sum_{\mathbb{A}}$ to $\int_{\mathbb{z}}$.
>
> As you have mentioned, if the auxiliary (or latent) random variables $\mathbb{A}$ is eliminated through marginalization, the system loses observer independence. On the other hand, if we do not eliminate it, the current probability theory stops at step (b). (Of course, approximation methods or analytical solutions for special datasets can handle it further.)
>
> **Q3**: a situation where all randomness in the world is governed by an abstract space $\Theta$, where all randomness is lost if the abstract space is determined at a point $\theta \in \Theta$ , and where all variables can be described deterministically.
>
> A3: We like your point and also such kind of high-level discussion. However, we are concerned that the point may have problem at the microscopic level according to 'quantum uncertainty'.
>
> Our further opinion is that the world can be very good predicted if we have sufficient multivariate latent variables $\mathbb{A}$ (or $\mathbb{z}$).
> This can be expressed as:
> $$P^{\mathbb{A}}\left ( Y\mid x_{n+1}   \right ) $$
>
> Where $\mathbb{A}$ is latent variables and $Y$ is our interested variables. By utilizing a sufficient number of historical observations of $\mathbb{A}$ adn $Y$, we can use this equation to make a prediction. Although this calculation is very complex, it is of polynomial complexity rather than exponential. Additionally,  this theory can be applied to time series forecasting.
>
> **Q4**: In the abstract space, random variables are represented as a projection of the world as a map to an object, e.g. $Y(\theta),X(\theta),A(\theta)$, can be uniquely determined for a given source $\theta$.
>
> A4: Firstly, $X$ is a very special random variable, conditioning on $X$ is only used to indicate which random experiment is being referred to.
> We'd like to express the probability space as abstract variables $\mathbb{A}$ (or $\mathbb{z}$) and $Y$, and $P_{\theta}(\mathbb{A}|x_{k})$ and $P_{\theta}(Y|x_{k})$ can be uniquely determined for a given source $\theta$.
> Our applications of IPNN, CIPNN and CIPAE serve as good illustrations of this point.
>
> **Q5**: Not your question but you may have interest. Why the auxillary observers are general?
>
> A5: The world is understood through observers. In the case of a coin toss, the outcome cannot be known unless observed. That is, the ground truth is not able to be known, we only know the observations.
>
> Given that the use of observers is unavoidable, and the imperfect observations maybe more general in the real world? (Perfect observations are the special case.)
>
> Additionally, we do not need to restrict the observers, as they may have different ways of understanding the world. For example, in our coin toss example in Appendix B, Observer$_3$ understands the outcome as a Gaussian distribution.
>
> The questions now is: what truly matters?
>
> The Change! While different observers may have different understandings of the world, the Change still follows the Ground Truth (albeit with some errors). Our proposed equation is designed to evaluate this Change.
>
>
>
> Finally, we are willing to have more discussion with you and look forward to your feedback.
>
>
> Best regards
>
> Authors

---

> > ### Comment · Reviewer_MWjN · 2023-11-17
> > **A quick thank you for authors' response**
> >
> > I appreciate the authors' very enlightening response. I am getting a much more correct understanding of the value of this paper thanks to the authors' response. In particular, A2 is very clear in its phrasing, which expresses the new ideas of this paper in a straightforward manner, and A5 does a good job of explaining the novelty of the paper conceptually.
> >
> > I believe I should raise my rating for this paper. To that end, I would like to take some time this weekend to read the paper again. I am contacting the authors first only to thank you, to let you know that I have indeed received and read the authors' response.

---

> > > ### Author Response · Authors · 2023-11-17
> > > **Thank you very much for your patience and efforts**
> > >
> > > Actually, this is already our third submission. This theory seems to be easily overlooked due to its simple mathematics and 'confusing concepts', we really appreciate your patience and efforts.

---

> > > > ### Comment · Reviewer_MWjN · 2023-11-19
> > > > **Update after rereading the manuscript**
> > > >
> > > > I have once again read the paper over with reference to the author's response and the comments of the other reviewers. Once again, I thank the authors for their kind responses and the reviewers for their important comments.
> > > >
> > > > In conclusion, my thoughts on this paper have been updated as follows:
> > > > - *Does this paper offer new and interesting ideas?*
> > > >
> > > > Without a doubt. I am convinced that this paper provides a new outlook and perspective on how to handle posterior probabilities. I think the ideas are worth sharing with the community and discussing often.
> > > > - *Does this paper build a new probability theory?*
> > > >
> > > > I believe that the manuscript, at least in its current form, is inadequate as a systematic theory and has room for major improvement.
> > > >
> > > > These impressions are due to my subjective perceptions as follows:
> > > > - Ideas are easy to understand when they are explained using concrete examples or special cases.
> > > > - On the other hand, I expect that a theory should be a generalization of a core idea.
> > > >
> > > > From this point of view, I think this paper is a very important contribution in explaining that new idea. So, is there sufficient generality to make it a new theory compared to the long-established probability theory (which is a very general theory with a very high degree of freedom)? I am unable to read sufficient generality into the manuscript at this time.
> > > > One important example, as other reviewers have pointed out, is whether the author's idea would work successfully in a measure-theoretic probability theory where classical (fractional form) conditioning does not work (a situation where conditioning is expressed by conditional expectation). If it works, does it require any new assumptions other than axioms 1-3? Another example (again, an extension of the measure-theoretic probability perspective) is whether it can be defined using an abstract probability space, as in my first comment.
> > > > If this paper provides a sufficiently generalized theory, one would expect to find these simple questions resolved, for example. These are only a few examples, and I suspect that the actual theory would require an enormous amount of work (especially if the opponent is probability theory, which has a long history).
> > > >
> > > > In summary, I expect that it would be very useful for the community to narrow the scope of this paper and present it with the granularity of new ideas, perspectives, and prospects. On the other hand, if this paper is presented as an innovative theory-building paper, I suspect that it will need to be generalized and comprehensive. And if the latter is the place to present an innovative theory-building paper, I think a journal with more mathematicians, statisticians, and theorists than a major machine learning conference would be more appropriate.
> > > >
> > > > Once again, I would like to thank the authors for sharing their great ideas and for their thoughtful responses. I hope this research is published where it deserves to be and that many more people will learn of its importance in the near future.

---

> > > > > ### Author Response · Authors · 2023-11-20
> > > > > **2nd Response to Reviewer MWjN**
> > > > >
> > > > > Dear Reviewer MWjN,
> > > > >
> > > > > Thank you very much for your time and detailed response, from your replying we can see that you tried a lot for helping us clarifying our paper in detail. We are very grateful for your efforts, especially for using your weekend time.
> > > > >
> > > > >
> > > > > Best regards
> > > > >
> > > > > Authors

---

### Author Response · Authors · 2023-11-17
**Just find one toy counterexample to deny the three new independent candidate axioms!**

Dear Reviewers,

We sincerely thank all of you for taking the time to read our paper.

In order to understand the value of this paper, we would like to make the following suggestions:

1. We strongly recommend that you review the simple coin toss example provided in Appendix B. If you agree that such a straightforward problem cannot be solved analytically using existing probability theories, then it should be clear that our proposed theory is indeed a new one. If we cannot come to a consensus on this point, then it may be best to reject the paper without further consideration.

2. Our proposed equation is derived from very fundamental probability principles: the independent equation, the total probability equation and the marginalization equation. It is rare for new theories to arise from such basic knowledge, and as such, we believe that it is worth examining our paper in detail.

3. Assuming that you agree, we suggest focusing on the three candidate axioms that we have proposed. Axioms are a crucial concept, and should not be accepted without careful examination.
Although we have conducted tests to support these axioms, it is never enough to validate a new axiom. However, to deny a new axiom, all you need is to find a single exception, even a simple toy example would suffice. (For example, the independent assumption of naïve Bayes theorem is strong, and it is easy to find one toy counterexample.)

4. If, after thorough investigation, you are unable to find any counterexamples, we encourage you to take these axioms seriously. They are not merely independent assumptions, but rather integral components of our proposed theory. This is why we assert that our proposed theory is a general analytical equation.


Finally, with this equation, we are able to easily and analytically handle a thousand (or more) dimensional continuous latent space.

Best regards

Authors

---

### Meta-Review · Area_Chair_m2Qm · 2023-12-11

**Metareview:**

This submission claims that there is no mathematical analytical form for a general posterior and that they have discovered a new theory to address this issue, which is called Indeterminate Probability Theory. The submission is unclear in too many aspects. The conclusion is that we cannot fully appreciate the ideas and results (assuming there is something to appreciate) because of the imprecise presentation of very core theoretical matters. This needs to be done in a very precise matter building up on literature.

**Justification For Why Not Higher Score:**

Presentation is not to the level that can be fully understood by members of a short-window reviewing process. Topic is very basic/core and likely should seek a more suitable venue for it.

**Justification For Why Not Lower Score:**

N/A

---

### Decision · Program_Chairs · 2024-01-16

Reject